# Deep learning modeling m⁶A deposition reveals the importance of downstream *cis*-element sequences

Zhiyuan Luo[1], Jiacheng Zhang [2], Jingyi Fei [3,4] & Shengdong Ke [1✉]

The $N^6$-methyladenosine (m⁶A) modification is deposited to nascent transcripts on chromatin, but its site-specificity mechanism is mostly unknown. Here we model the m⁶A deposition to pre-mRNA by iM6A (intelligent m⁶A), a deep learning method, demonstrating that the site-specific m⁶A methylation is primarily determined by the flanking nucleotide sequences. iM6A accurately models the m⁶A deposition (AUROC = 0.99) and uncovers surprisingly that the *cis*-elements regulating the m⁶A deposition preferentially reside within the 50 nt downstream of the m⁶A sites. The m⁶A enhancers mostly include part of the RRACH motif and the m⁶A silencers generally contain CG/GT/CT motifs. Our finding is supported by both independent experimental validations and evolutionary conservation. Moreover, our work provides evidences that mutations resulting in synonymous codons can affect the m⁶A deposition and the TGA stop codon favors m⁶A deposition nearby. Our iM6A deep learning modeling enables fast paced biological discovery which would be cost-prohibitive and unpractical with traditional experimental approaches, and uncovers a key *cis*-regulatory mechanism for m⁶A site-specific deposition.

[1] The Jackson Laboratory, Bar Harbor, ME 04609, USA. [2] Graduate Program in Biophysical Sciences, The University of Chicago, Chicago, IL 60637, USA. [3] Department of Biochemistry and Molecular Biology, The University of Chicago, Chicago, IL 60637, USA. [4] Institute for Biophysical Dynamics, The University of Chicago, Chicago, IL 60637, USA. ✉email: kelab018@gmail.com

The $N^6$-methyladenosine (m6A) modification is the most common internal modification in eukaryotic mRNA, and widely distributed in various tissues[1,2]. It was first identified to be in mRNA during 1970s[3–5]. m6A is involved in diverse biological processes including cell differentiation, cancer progression and neurological development[6–10]. Due to its functional importance, m6A has been discovered to affect various aspects of RNA biology, including splicing, polyadenylation, export, degradation, and translation[11,12]. Its major function is believed to regulate mRNA turnover[13–16].

The m6A modification on mRNA is catalyzed by the m6A methyltransferase complex (MTC), which is comprised of METTL3 and METTL14 as the catalytic core[17–19]. Additional components including WTAP, VIRMA, ZC3H13, and HAKAI are also found to interact with METTL3-METTL14 and affect the complex activity[20–26]. The m6A consensus sequence RRACH as a stringent motif or RAC as a more inclusive motif (R = A or G, H = A, C, or U) was first determined by biochemical experiments[27–30]. Despite the wide prevalence of the m6A consensus motif in transcripts, very few of them are methylated, highlighting the site-specificity of m6A methylation. To investigate the global m6A distribution at the transcriptomic level, the m6A-seq and MeRIP-seq were first developed to map the m6A peak regions (typically ~200 nt or longer) using commercially available m6A antibodies[1,2]. To achieve single-nucleotide resolution mapping for m6A, m6A-CLIP/miCLIP/PA-m6A-seq cross-linked m6A-antibody to its m6A mRNA target by UV and achieved precise m6A mapping in transcripts by detecting the reverse transcription errors due to the residual peptide crosslinked to m6A (CIMS sites: crosslink induced mutational sites, CITS sites: crosslink induced truncation sites)[31–33]. In addition, the reverse transcription errors by m6A modification itself also enabled the single-nucleotide resolution mapping (MITS sites: m6A induced truncation sites)[33]. Though a few new precise m6A mapping methods have recently been developed by exploring alternative ideas[34–38], the m6A-CLIP method has generated the major share of precise m6A sites in human and mouse transcripts[8,14,32,33,39]. The m6A mapping studies showed that m6As were preferentially enriched in last exons, both their coding region and 3′UTR (untranslated region), as well as in long internal exons[1,2].

Based on the existing m6A sites precisely determined by experiments, computational methods have been developed to model the m6A sites in mRNA, including the machine learning-based methods (WHISTLE, SRAMP, and MethyRNA) and the deep learning-based methods (TDm6A, DeepM6ASeq)[39–43]. These bioinformatics methods mostly focused on the gradually improvement for the m6A site modeling accuracy, but used the relatively small-scale data integration and contributed little to discovery of biological mechanisms.

Here we first described a new deep learning method, the ResNet (residual neural network), for modeling the m6A deposition in pre-mRNA. The ResNet avoids the vanishing gradient problem in deep neural networks by the skip connections[44]. Skip connections allow to skip some layers in the neural network and feed the output of one layer as the input to the next layers[44], enabling us to build deeper neural networks (adding more layers) and improve the accuracy of classification. In addition, it can handle very large datasets to investigate more complex issues. This deep learning method has been successfully used to handle high-throughput sequencing data and model biological processes[45,46].

Our ResNet deep learning approach, the iM6A (intelligent m6A), models the m6A site-specific deposition in the genome with a state of art accuracy. Using saturated mutational analysis to generate input sequence, we systematically perturbed the input sequence to the iM6A deep learning model to see how it affects the m6A deposition output. We discovered surprisingly that the downstream 50 nt region of the m6A sites contained a high density of the cis-elements for the m6A deposition. This pattern was consistently true for both last exons and internal exons. We further characterized m6A enhancers and silencers by implementing linear regression to interpret the iM6A deep learning output. The iM6A modeling as well as the identified functional cis-elements were validated by independent experimental data and evolutionary conservation. By a similar process of model perturbation, we found that synonymous codon mutations can affect m6A deposition and that the TGA stop codon may promote the adjacent m6A deposition. The iM6A approach enabled high-throughput and effective biological discovery which would be cost-prohibitive for traditional experimental methods, and uncovered a key cis-regulatory mechanism governing m6A site-specific deposition.

## Results

**iM6A accurately models m6A deposition.** As with any nucleotide-related biological process, the question arises whether the site-specificity of m6A deposition is determined in whole or part by a "code" in flanking primary nucleotide sequences. Is there an m6A cis-element code? To address this question directly, we developed the iM6A (intelligent m6A, Fig. 1a), a deep residual neural network (ResNet)[44] to model the m6A site-specific deposition at genome-wide level. We first collected a high-quality set of m6A sites that were precisely determined by the m6A-CLIP experiments in mouse transcriptome[14,33]. We used pre-mRNA sequences as input: the m6A sites on pre-mRNA were served as positive sites, while the remained nucleotides were treated as negative sites. The whole dataset was divided into training and test datasets. The training dataset contained all the transcripts on most chromosomes except chromosome 9 (Chr9), the transcripts of which were held out and reserved for the later independent test of iM6A modeling. iM6A evaluated the full length of transcripts, and the outputs of which were probabilities of each nucleotide position being an m6A site (see details in the Methods). iM6A modeled the m6A sites in the test set with an accuracy of 0.991 as measured by the AUROC score (area under receiver operator curves) (Fig. 1b). As the comparison, we also implemented a traditional machine learning method, SVM (Support Vector Machine)[39] and an alternative deep learning method CNN-RNN (Convolutional Neural Network-Recurrent Neural Network)[40] to modeling the m6A modification deposition for the same training and testing datasets (see Methods for more details). The comparisons showed that iM6A achieved better performance than both SVM and CNN-RNN (Fig. 1b).

Alternatively, the performance of iM6A measured by the AUPRC score (area under precision recall curves) showed iM6A was also better than those of SVM and CNN-RNN methods (Supplementary Fig. 1b). The m6A sites experimentally determined by m6A-CLIP were accurately identified from the non-methylated sites by iM6A in the independent test (Fig. 1c). For comparison with mouse, we implemented the iM6A strategy to model the m6A site-specificity in human genome by using a high-quality set of human m6A sites that were precisely determined by the m6A-CLIP experiment[8,14,32,33] and obtained the same high AUROC and AUPRC performances (Supplementary Fig. 1a, c, d).

Our iM6A training was using the experimentally determined m6A sites by the m6A-CLIP method which identified a major share of the single-nucleotide resolution m6A sites that had been mapped so far (Supplementary Fig. 1l). To make sure that the iM6A model was accurate for all m6A sites independent of the experimental methods that precisely mapped them, we examined

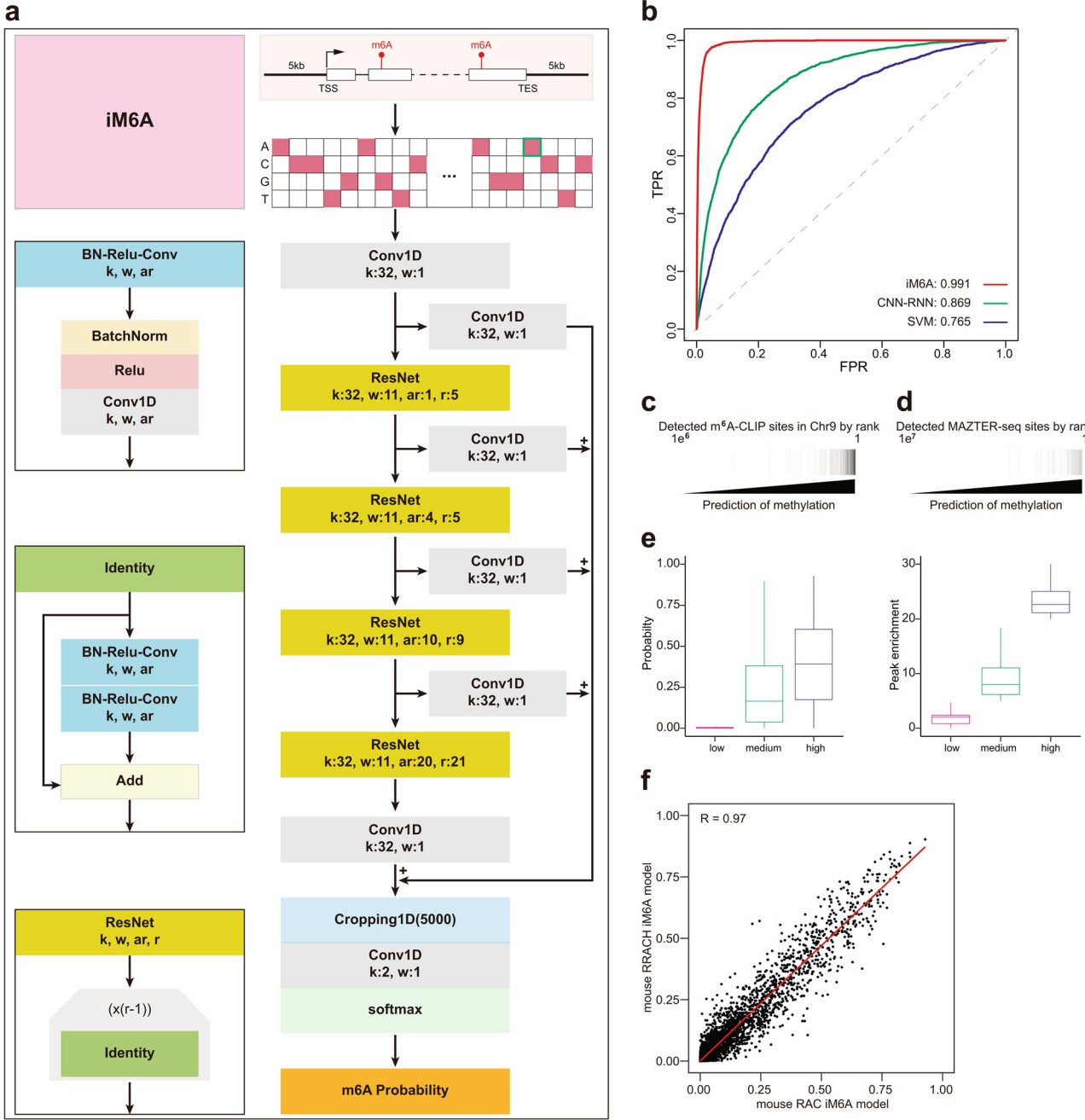

**Fig. 1 iM6A models m⁶A deposition with single-nucleotide resolution. a** iM6A architecture. The architecture started with a convolutional layer (Conv), then was followed by four Residual Network (ResNet) blocks, where k, w, ar, and r are the number of convolutional kernels, window size, dilation rate of each convolutional kernel in the layer, and repetition numbers respectively. Further, the output of every ResNet block was added to the input of penultimate layer, connected with a convolutional unit with softmax activation. **b** Receiver operator curves (ROCs) and corresponding area under receiver operator curves (AUROC) scores of iM6A, CNN-RNN (implemented in TDm6A[40]), and SVM (implemented in MethyRNA[43]). Here mouse chromosome 9 data was used to test the iM6A, CNN-RNN, and SVM models, which were trained independently on data of other mouse chromosomes except chromosome 9 (similar result was obtained for human m⁶A modeling in Supplementary Fig. 1a). **c** Heatmap of the iM6A modeling and m⁶A-CLIP detected sites in mouse chromosome 9. The modeled sites were sorted based on modeled score, the black line denoted whether methylation was identified at the sites by m⁶A-CLIP[14,33]. **d** Heatmap of the iM6A modeling and MAZTER-seq detected sites. The modeled sites (conformed to RRACA) were sorted based on modeled score, and the black line denoted whether methylation was identified at the sites by MAZTER-seq[34]. **e** The modeled probability by iM6A agreed with the experimentally quantified m⁶A methylation level. Modeled probability by iM6A (left panel) and enrichment score quantified by m⁶A-CLIP (right panel) at mouse m⁶A sites. The m⁶A sites were categorized into three groups based on their m⁶A peak enrichment value as the low (n = 63,0854), medium (n = 76,111), and high (n = 3229) groups. Median and interquartile ranges are presented for the box plot. **f** Scatter plot of modeled probability for m⁶A sites (n = 100,000) in mouse chromosome 9 using mouse RAC iM6A model versus mouse RRACH iM6A model. Each dot represented one site in mouse chromosome 9 discovered by both models, and the labeled axes provided the probability values for that site by the two models.

whether iM6A could identify m6A sites mapped by alternative experimental methods (Supplementary Fig. 1g) including m6A-label-seq[36], MAZTER-seq[34], m6ACE-seq[35], and miCLIP2[47]. The m6A-label-seq method detected m6A sites by chemically substituting the m6A with a6A ($N^6$-allyladenosine) at the m6A sites, MAZTER-seq identified a relatively small subset of m6A sites that were in the m6ACA motifs by a methyl-sensitive RNase, and m6ACE-seq detected m6A sites by its crosslinking to the m6A-antibody and followed with the exonuclease digest to achieve single-base resolution. In addition, miCLIP2 was an optimized CLIP method that combined miCLIP with machine learning to improve m6A detection[47]. The precisely mapped m6A sites by all these alternative experimental methods were identified with high probability values by iM6A (Fig. 1d and Supplementary Fig. 1j, k for mouse, Supplementary Fig. 1e–i for human), indicating that iM6A modeling was accurate and supported by a variety of the m6A mapping experimental methods. Furthermore, we investigated if the modeled m6A probability by iM6A for an m6A site was quantitatively associated with its methylation level. The m6A peak enrichment value quantifies its methylation level by normalizing the m6A-IP read count to the input read count for an m6A peak region[1,2,14,33]. We categorized the m6A sites into three groups based on their m6A peak enrichment value as the low, medium, and high groups, and found that the modeled m6A probability by iM6A associated with the quantitative distribution of the peak enrichment value across the three groups (Fig. 1e for mouse, and Supplementary Fig. 1m for human). MAZTER-seq is another method that could experimentally quantify m6A methylation level for a small subset of m6A sites that were in RRACA (R = A or G) motif, with the higher m6A methylation level associated with the lower cleavage efficiency by a methylation-sensitive RNase[34]. The modeled m6A probability by iM6A also associated with the quantitative distributions across the different cleavage efficiencies groups (Supplementary Fig. 1n for mouse data, and Supplementary Fig. 1o for human data). All the results above supported that the m6A probability score generated by iM6A reflected quantitatively the methylation level at the m6A site.

It is known that the m6A site consensus could be either RRACH (H = A, C, or U) as a high stringent set or RAC as a more inclusive set. Accordingly, to be comprehensive, we independently trained the RAC iM6A model and the RRACH iM6A model using either the RAC or the RRACH experimentally determined m6A sites in most genes on chromosomes except chromosome 9 (Chr9) as the training dataset, and tested the performance of the RAC and the RRACH iM6A models on genes from chromosome 9 (Chr9). The RAC iM6A model performed very similarly to the RRACH iM6A model (Fig. 1f for mouse data, and Supplementary Fig. 1p for human data). In addition, we trained the iM6A model with 80, 400, 2 K, and 10 K sequence on both sides, and the performance increased along with sequence length (Supplementary Fig. 1q). For all the analysis in the remaining result section, we implemented the RAC iM6A-10K model to generate all the data.

**Cis-elements that govern the m6A deposition locate largely within 50 nt downstream of the m6A sites**. Though iM6A as a deep learning approach was powerful in accurately modeling the m6A sites in the genome, this deep learning black box did not aid understanding of the underlying *cis*-element rules, i.e., the m6A *cis*-element code. To systematically identify the *cis*-elements that determine m6A modification, we performed single nucleotide saturation mutagenesis (Fig. 2a) to the sequences flanking the m6A sites in last exon which contains about 70% of all m6A sites in the transcripts[33] and calculated the positional mutational

effects for the m6A deposition by iM6A. We found that the mutations that either increased or decreased m6A probability significantly ($|\Delta \text{Probability}| > 0.1$) were largely enriched in the downstream region of the m6A sites, especially within the 50 nt downstream of m6A sites (Fig. 2b for mouse; and Supplementary Fig. 2a for human), suggesting *cis*-elements that influence m6A deposition locate largely in this region.

While the last exon hosts a majority of m6A sites, the long internal exon also contains many m6A sites[14]. We applied the same strategy to investigate the *cis*-elements flanking the m6A sites in the long internal exon, and found that the downstream 50 nt region of the m6A sites again contained largely of the *cis*-elements that regulate the m6A deposition (Fig. 2c for mouse; and Supplementary Fig. 3b for human), suggesting that the m6A deposition in both the last exon and the long internal exon may follow a similar mechanism.

To systematically and quantitatively analyze the *cis*-element effect on m6A deposition in the 50 nt downstream region, we implemented a linear regression approach (Fig. 2d) which had been demonstrated to be effective in identifying functional motifs for microRNA targeting[48] and pre-mRNA splicing regulation[49]: a substitution was made which created and disrupted five overlapping 5-mers simultaneously and the net effect for each pentamer motif was determined by the slope of the linear regression equation when pooling all the data (see details in the Methods). Based on their effect value and the statistical significance, pentamer motifs were ranked, Top 20 enhancers and silencers were showed. For the last exon, the m6A enhancers included mostly part of the RRACH motif; the m6A silencers mostly contained the CG/GT/CT dinucleotides (Fig. 2e, f for mouse, and the virtually the same motif set for human, Supplementary Fig. 2c, d). Almost the same set of the m6A enhancers and the m6A silencers were obtained for the long internal exon (Supplementary Fig. 2f, g). We also observed a strong effect value correlation for all pentamers between the study in the last exon and the study in the long internal exon (Fig. 2g for mouse, and Supplementary Fig. 2e for human), supporting that the same *cis*-element code governed m6A site-specific deposition in both locations. Moreover, the strong effect correlation was obtained for all pentamers between the study in mouse and the study in human (Supplementary Fig. 2h for the last exon, and Supplementary Fig. 2i for the long internal exon), supporting that both mouse and human had the same *cis*-element code in regulating m6A deposition for both the last exon and the long internal exon.

We further investigated the m6A enhancer and silencer motif distribution in the region flanking the m6A sites. The m6A enhancers had a higher frequency around the m6A sites than the control that had the exact RAC motif matched (Supplementary Fig. 2h for mouse, and Supplementary Fig. 2j for human). In contrast, the m6A silencers had a lower frequency around the m6A sites than the control that had the exact RAC motif matched (Fig. 2i for mouse, and Supplementary Fig. 2k for human). The difference in the downstream region of the m6A sites was more evident than upstream region (Fig. 2h, i for mouse, Supplementary Fig. 2j, k for human), supporting the hypothesis that the functional *cis*-elements largely resided in the 50 nt downstream of the m6A sites. Next, we examined the m6A enhancer and silencer motif distribution on several sets of the experimentally mapped m6A sites by different methods, including m6A-CLIP, m6A-label-seq, m6ACE-seq, and MAZTER-seq. The m6A enhancers showed consistently higher frequency in the positive m6A sites than the control (Supplementary Fig. 3a, c, e, g, i, k, m) for both human and mouse dataset, while the m6A silencers exhibited lower frequency (Supplementary Fig. 3b, d, f, h, j, l, n). All of these positional distribution investigations confirmed that the

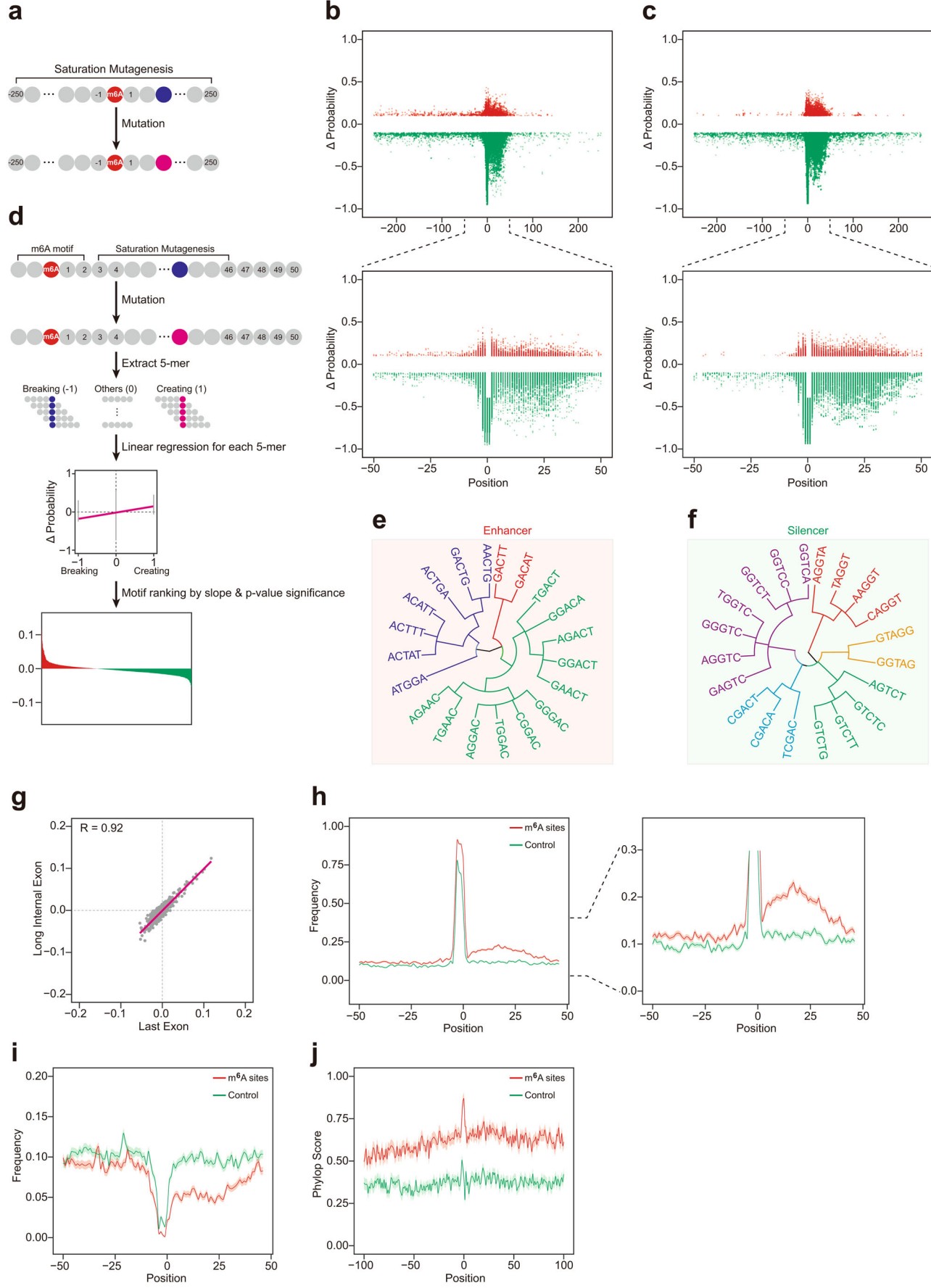

**Fig. 2 *Cis*-elements that regulate m⁶A deposition locate largely within 50 nt downstream of m⁶A sites. a** Evaluate positional mutational effects to m⁶A deposition by single nucleotide saturation mutagenesis. For each site in the sequences (−250 to 250) around the m⁶A site, the nucleotide was substituted by each of three other nucleotides. The delta changes of m⁶A probability value (ΔProbability) after mutation was calculated according to iM6A.
**b, c** Positional plot of ΔProbability (cutoff = 0.1) for the m⁶A sites located in last exon (**b**) or long internal exon (**c**). Up panel: dot plot of ΔProbability for the sequences (−250 to 250) around the m⁶A site. Bottom panel: dot plot of ΔProbability for the sequences (−50 to 50) around the m⁶A site. Red color dots were mutational events that increased m⁶A probability; Green color dots were mutational events that decreased m⁶A probability. **d** The systematic and quantitatively determination of the effect on m⁶A deposition for all *cis*-element pentamers by linear regression (See details in Methods). Total number of m⁶A sites used was 1500. (The full list of effect values for each pentamer motif are listed in Supplementary Data 2). **e, f** Dendrogram showed clustering of Top 20 enhancer and silencer motifs. Enhancers mostly contained part of RRACH motif, and silencers mostly contained CG/GT/CT motifs. **g** Scatter plot for the effect correlation for all pentamers between the study in last exon and the study in long internal exon. The effect of each pentamer motif was determined by the slope of linear regression equation, and each gray dot was a pentamer. **h–j** Positional plot for the frequency of Top 100 enhancers (**h**), silencers (**i**), and conservation scores (**j**) in the sequences around the m⁶A sites. The plots were compared between the higher m⁶A probability (red color, probability ≥ 0.7) and the lower m⁶A probability (control, green color, probability < 0.1) of RAC sites. Data were presented as mean ± S.E.M. standard error of the mean.

frequency difference for the m⁶A enhancers and silencers was more evident in the downstream region of m⁶A sites than the upstream region, and generally true regardless of the experimental approaches that mapped the m⁶A sites.

Furthermore, we conducted the study for the sequence conservation flanking the m⁶A sites and found that the flanking sequences of the m⁶A sites were more conserved than that of the control (Fig. 2j for mouse, and Supplementary Fig. 2l for human). Moreover, the functionally greater importance of the downstream sequences flanking m⁶A sites compared with upstream sequences was supported by their being more conserved cross species and the fact that such conservation did not exist in the control. (Fig. 2j for mouse, and Supplementary Fig. 2l for human).

At last, given that the enhancers surrounding the m⁶A sites include mostly part of RRACH motif, which are potential motif for methylation, we examined the distribution of methylated sites flanking the m⁶A sites. We found that the RAC sites adjacent to m⁶A sites have a higher frequency to be m⁶A sites (Supplementary Fig. 2m), indicating it's more likely to be methylated. The RAC sites adjacent to non-m⁶A sites have lower frequency to be m⁶A sites (Extended Data Fig. 2n), indicating it's unlikely to be methylated. Moreover, both methylated and non-methylated RAC sites are enriched in the downstream 50 nt region of m⁶A site (Extended Data Fig. 2m), suggesting both could enhance m⁶A deposition.

Taken together, our data strongly supported that the *cis*-elements regulating m⁶A deposition largely reside within the 50 nt downstream of the m⁶A sites, with additional functional subsequences being less concentrated in other regions (Fig. 2b, c, and Supplementary Fig. 1q). Enhancers include mostly part of RRACH motif, while silencers generally contain CG/GT/CT motifs.

**Experimental validation of the iM6A modeling**. By an independent experimental dataset, we validated the m⁶A deposition modeling by iM6A. The lymphoblastoid cell lines (LCLs) were from a collection of 60 Yoruba (YRI) human individuals. m⁶A signals of LCLs were experimentally determined by m⁶A RIP-seq method (m⁶A RNA immunoprecipitation and sequencing) in the transcriptome[50]. Within the genomes of the 60 individuals, there was adequate data to obtain a reference allele, alternate SNVs (single-nucleotide variants), and heterozygote examples. It was now possible to investigate how SNVs influence m⁶A deposition. We implemented a computational method (see Fig. 3a and Methods section) to quantify the association between a specific SNV and the m⁶A level of an m⁶A peak region in which this SNV located. The m⁶A peak regions that contained the m⁶A sites were usually 200 nt or longer. iM6A calculated the effect of specific SNVs on m⁶A deposition and identified 47 SNVs that either

increased or decreased the m⁶A deposition (|ΔProbability| > 0.1). Among them, the statistical majority (33 SNVs out of 47, *P* < 0.004, Binomial test, Fig. 3b) had the same directional change in m⁶A deposition modeled by iM6A and as determined experimentally. Furthermore, we examined the value correlation between the iM6A modeled m⁶A deposition changes (ΔProbability) and the experimentally measured m⁶A deposition changes (ΔPeakEnrichment), and found that iM6A quantitatively modeled the experimental m⁶A deposition changes (Fig. 3c, *P* < 0.0003, Student's *t*-test). Among the 47 SNVs, ten located upstream of m⁶A sites, four were at the m⁶A site, and the remaining 33 located within the downstream 50 nt of m⁶A sites (Fig. 3d). Thus, SNVs that affected m⁶A deposition were statistical biased towards downstream (*P* < 0.0002, Binomial test), supporting that the downstream region of m⁶A sites contained largely the *cis*-elements regulating m⁶A deposition. We also found that the underlying *cis*-elements alterations for each SNV represented as the sum of the effective value changes for all the involved pentamers (ten pentamers in total, five pentamers disrupted and created simultaneously) quantitively agreed with the experimental m⁶A deposition changes (Fig. 3e, *P* < 0.0001, Student's *t*-test). Four examples in which the m⁶A deposition was affected by an SNV were shown in Fig. 3f–i. The rs7831 in *PDCD11* gene was an A to C mutation that was modeled by iM6A to be at an m⁶A sites and decrease its m⁶A probability value from 0.8 to about 0, and indeed we observed an evident loss of experimental m⁶A signal in the alternative allele data (C nucleotide, blue color) in comparison to that of the reference allele data (A nucleotide, red color) (Fig. 3f). The rs75907001 in *DOPEY2* gene was a T to C mutation which was modeled by iM6A to decrease the m⁶A probability value of the m⁶A site from 0.6 to about 0.4, and an experimental m⁶A signal decrease was observed (Fig. 3g). This T to C mutation located 4 nt downstream of an m⁶A site, disrupted three m⁶A enhancer motifs (ACTCT, CTTGG, and TTGGG), and simultaneously created four CG/GT/CT-containing silencer motifs (ACTCC, CTCCT, TCCTG, and CCTGG) (Fig. 3g). The rs9090 in *PARM1* gene was a C to T mutation which was modeled by iM6A to increase the probability value of the m⁶A site from about 0 to 0.15, and an experimental m⁶A signal increase was recorded (Fig. 3h). This C to T mutation located three nucleotides downstream of an m⁶A sites, created four enhancers (AGACT, GACTG, ACTGT, and CTGTT), and one silencer (TGTTT) and simultaneously disrupted three silencers (GACCG, ACCGT, and CCGTT), leading to an overall increase of m⁶A signal supported by both the iM6A modeling and the experimental data (Fig. 3h). Another example was the rs1057278 located in *TTLL3* genes. This G to A mutation located eleven nucleotides downstream of an m⁶A sites according to iM6A, disrupted four CG/GT/CT silencers (CAGGA, AGGGC,

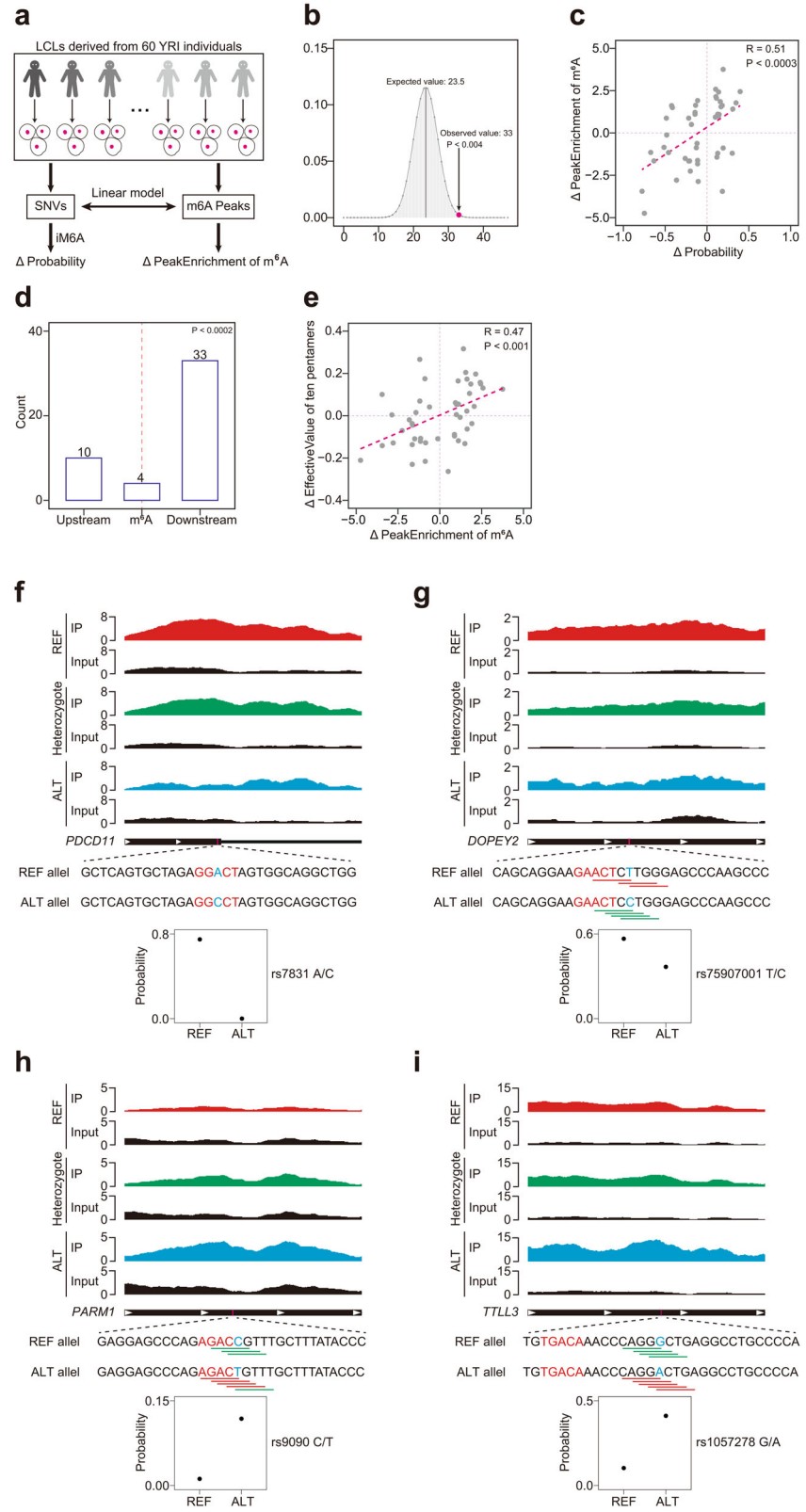

GGGCT, and GGCTG) and simultaneously created five enhancers (CAGGA, AGGAC, GGACT, GACTG, and ACTGA), collectively contributing to an increase of m⁶A signal again supported by both the iM6A modeling and the experimental data (Fig. 3i). Altogether, the experimental data supported the m⁶A deposition modeling by iM6A and that confirmed that the m⁶A regulating *cis*-elements locate downstream of the m⁶A sites.

**Many pathogenic SNVs are associated with m⁶A deposition changes.** Even though a number of studies have revealed that the dysregulation of m⁶A impacts various diseases[51], little is known about how nucleotide variants impact m⁶A deposition. To address this question, 68286 SNVs were extracted from the ClinVar database (https://www.ncbi.nlm.nih.gov/clinvar/) (see details in the Methods). As shown previously (Fig. 2),

**Fig. 3 Experimental validation of iM6A modeling. a** The workflow of the validation of iM6A by m⁶A-QTLs dataset (See details in Methods). **b** The change direction of experimentally measured m⁶A peak changes agreed with that of the iM6A modeling on the related SNVs. The expected value was 23.5 in random situation, and the observed value was 33. The *p*-value (<0.004) was calculated based on Binomial distribution. **c** The change quantity of experimentally measured m⁶A peak changes (ΔPeakEnrichment) agreed with that of the iM6A modeling on the related SNVs. The *R*-value was calculated by Pearson Correlation Coefficient, and *p*-value (<0.0003) was determined by two-sided Student's *t*-test. **d** Bar plot of SNVs distribution over the corresponding m⁶A site. 10, 4, and 33 SNVs were located upstream, at, or downstream of the corresponding m⁶A site respectively. The *p*-value (<0.0002) was calculated based on Binomial distribution. **e** The change quantity of pentamer effective value changes (ΔEffectiveValue) agreed with experimentally measured m⁶A peak changes (ΔPeakEnrichment) on the related SNVs. The *R*-value was calculated by Pearson Correlation Coefficient, and *p*-value (<0.001) was determined by two-sided Student's *t*-test. **f–i** Examples of SNVs affecting m⁶A deposition. rs7831 (Fig. 2f), rs75907001 (Fig. 2g) abolished or dampened m⁶A deposition respectively. rs9090 (Fig. 2h) and rs1057278 (Fig. 2i) enhanced m⁶A deposition. The m⁶A site was marked by red color, the mutation site was marked by blue color. The enhancers and silencers were marked by red and green lines, respectively. The m⁶A RIP-seq data for homozygote of the major allele, heterozygote, homozygote of the minor allele, and Input were marked by red, green, blue, and black color, respectively. The dot plot showed the probability value of the m⁶A with major allele or minor allele.

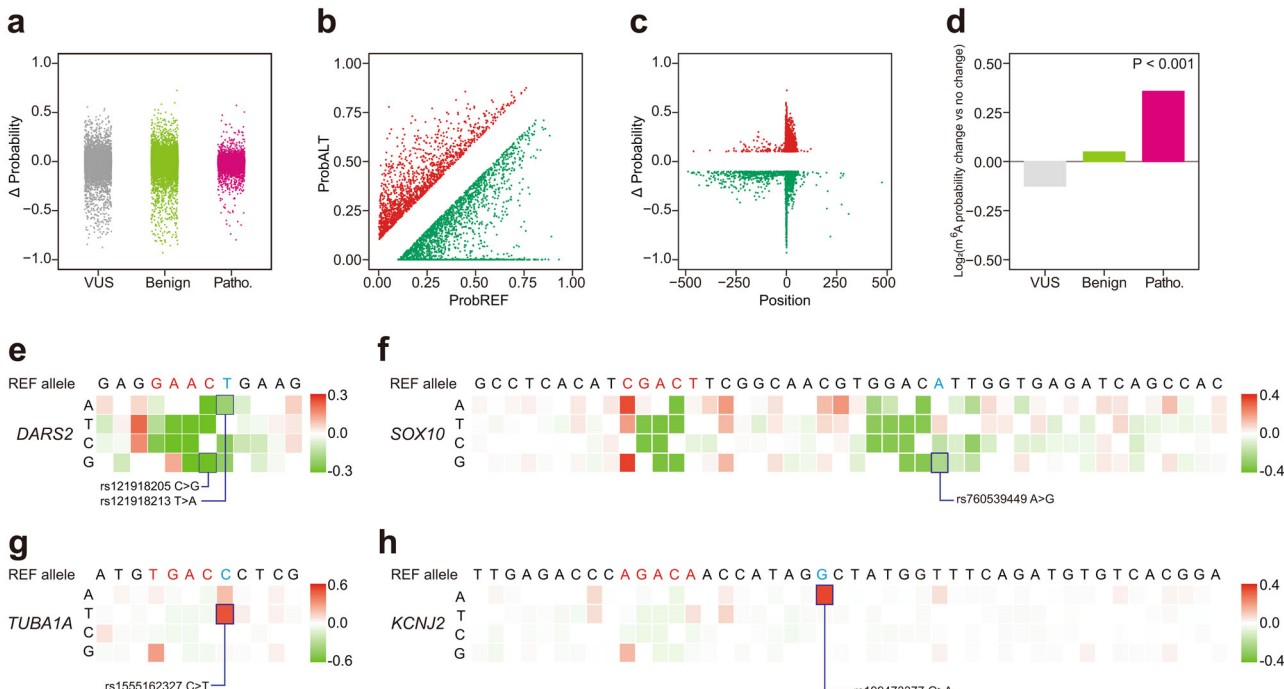

**Fig. 4 Many pathogenic SNVs are associated with m⁶A deposition changes. a** Strip plot of ΔProbability caused by SNVs, which were categorized by clinical significance in ClinVar. VUS means uncertain significance SNVs (gray), Benign means benign and likely benign SNVs (green), and Patho. means pathogenic and likely pathogenic SNVs (pink). **b** Scatter plot of modeled probability for m⁶A sites with major allele (ProbREF) or minor allele (ProbALT). Red color dots were mutational events that increased m⁶A probability (ΔProbability ≥ 0.1); Green color dots are mutational events that decreased m⁶A probability (ΔProbability ≤ −0.1). **c** Positional plot of ΔProbability (cutoff = 0.1) for m⁶A sites with major allele or minor allele. Red color dots were mutational events that increased m⁶A probability; Green color dots were mutational events that decreased m⁶A probability. Agreeing with the finding in Fig. 2, *cis*-elements that regulated m⁶A deposition located largely within 50 nt downstream of m⁶A sites. **d** Bar plot of log₂(odd ratio, m⁶A probability change group over no change group) of the percentage of SNVs with different clinical significances in ClinVar. The SNVs were categorized into two groups (m⁶A probability change or no change) based on ΔProbability (|ΔProbability| ≥ 0.1). The *p*-value (<0.001) was calculated by Fisher's exact test. **e–h** Saturation mutagenesis of the sequence flanking the m⁶A sites in *DARS2*, *SOX10*, *TUBA1A*, and *KCNJ2*. The heatmaps visualized ΔProbability of each mutation event, and were annotated with SNVs from ClinVar.

flanking sequence can influence m⁶A deposition. We selected the SNVs in the 500 nt region flanking the m⁶A sites. The clinical significance of these SNVs were categorized based on the ClinVar annotation. We found that many SNVs were modeled to alter m⁶A deposition, either enhancing or dampening (Fig. 4a). Though a large proportion of SNVs don't affect m⁶A deposition, some do have evident effects on m⁶A deposition (Fig. 4a). We focused on the events that could change the m⁶A probability (|ΔProbability| ≥ 0.1) (Fig. 4b), and found that many of these were also highly enriched in the region 50 nt downstream of the m⁶A sites (Fig. 4c), as was found previously for SNVs created by single nucleotide saturation mutagenesis (Fig. 2b, c,

and Supplementary Fig. 2a, b) and the SNV experimental validation data (Fig. 3d). To exclude the effect of SNVs on change of protein-coding sequence, we focused on the SNVs that only cause synonymous mutations, and these SNVs that could change the m⁶A probability (|ΔProbability| ≥ 0.1) were also highly enriched in the region 50 nt downstream of m⁶A sites (Supplementary Fig. 4a, b). We further categorized all the SNVs (68286 SNVs in ClinVar database) into two groups (m⁶A probability change or no change) based on ΔProbability (|ΔProbability| ≥ 0.1 or |ΔProbability| < 0.1), and we found that the pathogenic SNVs had greater prevalence in the group for which m⁶A probability significantly changed (*P* < 0.001, Fisher's exact test, Fig. 4d), demonstrating that

pathogenic SNVs were more likely to alter m$^6$A deposition than non-pathogenic SNVs.

Four examples in which m$^6$A deposition was affected by the pathogenic SNVs were shown in Fig. 4e–h. For rs121918205 and rs121918213 in *DARS2* gene, the rs121918205 was a C to G mutation that broke the C in the RAC consensus of an m$^6$A site, leading to a decreased m$^6$A probability value of −0.3 according to iM6A. The rs121918213 was a T to A mutation that led to a decreased m$^6$A probability value of −0.3 (Fig. 4e). The *DARS2* encodes mitochondrial aspartyl-tRNA synthetase, and its deficiency may be involved in leukoencephalopathy[52]. The two SNVs above could affect the m$^6$A modification in the *DARS2* transcript as a novel disease cause. The rs760539449 located in *SOX10* gene was an A to G mutation that led to a decrease m$^6$A probability value of −0.4 (Fig. 4f). The rs760539449 was annotated as likely pathogenic SNV in ClinVar. The iM6A modeled this mutation to result in a loss of the m$^6$A deposition as a potentially new disease mutation insight. The rs1555162327 in *TUBA1A* gene was a C to T mutation that generated an increased m$^6$A probability value of 0.6 according to iM6A (Fig. 4g). Another example was the rs199473377 located in *KCNJ2* gene, and this G to A mutation caused m$^6$A probability increase to 0.4 by iM6A (Fig. 4h). In summary, iM6A worked as a method to annotate the disease-related SNVs that could affect m$^6$A deposition. Even though all the SNVs showed in Fig. 4e–h also cause missense mutation, iM6A could help to annotate the effect of pathogenic SNVs on m$^6$A deposition beyond protein-coding sequence mutations. In ClinVar database, the missense and nonsense SNVs are more likely to be annotated as pathogenic for their convenience in inferring protein functional disruption. In other words, the pathogenic SNVs that are documented currently in ClinVar primarily focus on protein sequence disruption. Our iM6A annotation provides an alternative angle to interpret these disease-causing SNVs from the m$^6$A RNA modification perspective. As the m$^6$A disease research grows mature in the future, the ClinVar database could include pathological SNVs that was affected by m$^6$A deposition alone and our iM6A work could promote the disease research discovery in this direction. Defining the disease-associated mutations among millions of SNVs is a grand challenge. The database like RMvar[53], RMDisease[54] collected the genetic variants which might be associated with m$^6$A modification, while iM6A could provide synergistic contribution to decipher the *cis*-element mechanisms and could provide a new perspective in understanding the diseases caused by RNA modifications.

**Synonymous codons may influence m$^6$A deposition**. Since many m$^6$A sites locate in the coding region, an open hypothesis is whether synonymous codon usage affects m$^6$A deposition and serves as a new layer of regulation. To test this hypothesis, we performed saturation synonymous codon swap (Fig. 5a) for the coding sequences flanking the m$^6$A sites in the last exon and calculated with iM6A the positional mutational effects on m$^6$A deposition. We found the mutational events that either increased or decreased the m$^6$A probability significantly (|ΔProbability| > 0.1) were also highly enriched in the downstream region of the m$^6$A sites (Fig. 5b for mouse; and Supplementary Fig. 5a for human), supporting that synonymous codons that influenced m$^6$A deposition located in this region. Next, we systematically and quantitatively analyzed the effect of the synonymous codons on m$^6$A deposition in the 15 downstream codons (15 × 3 nt = 45 nt, covering the downstream 50 nt region). Similar to the pentamer analysis in Fig. 2d, we implemented the linear regression approach (Fig. 5c) to identify the effect of synonymous codon on m$^6$A deposition: each synonymous codon substitution created one codon and disrupted

the original codon simultaneously, the effect for each synonymous codon was determined by the slope of the linear regression equation. Based on their effect value and the statistical significance, synonymous codons were ranked for their effect in m$^6$A deposition. Top 10 enhancing or silencing synonymous codons and their corresponding amino acids were showed: the m$^6$A enhancing synonymous codons include mostly part of the RRACH motif; the m$^6$A silencing synonymous codons mostly have the CG/GT/CT motifs, agreeing with the pentamer motif property of the m$^6$A enhancers and the m$^6$A silencers (Fig. 5d for mouse, and the virtually the same codon set for human, Supplementary Fig. 5b). Interestingly, we saw that many sets of synonymous codons encoding the same amino acids contained codons with opposing effects on m$^6$A deposition. For example, both AGA and CGT encoded arginine (R) with the former enhancing m$^6$A deposition and the latter silencing m$^6$A deposition (Fig. 5e). More examples included the synonymous codon pair GAC and GAT for the aspartic acid (D) and the pair ACT and ACC for the threonine (T) (Fig. 5e). We also observed a strong effect correlation for all synonymous codons between the studies in mouse and in human, supporting that the same synonymous codon bias influenced m$^6$A site-specific deposition in both mouse and human (Fig. 5f). We further investigated the positional distribution of the m$^6$A enhancing and silencing synonymous codons in the region flanking the m$^6$A sites. The m$^6$A enhancing synonymous codons had a higher frequency around the m$^6$A sites than the control (Fig. 5g for mouse, and Supplementary Fig. 5c for human). In contrast, the m$^6$A silencing synonymous codons had a lower frequency around the m$^6$A sites than the control (Fig. 5h for mouse, and Supplementary Fig. 5d for human). The density difference for the enhancing/silencing synonymous codon is more evident in the downstream region of an m$^6$A site than its upstream region (Fig. 5g, h for mouse, and Supplementary Fig. 5c, d for human), arguing that the functional *cis*-elements fall more often in the 50 nt downstream of the m$^6$A sites.

**The stop codon TGA may favor the m$^6$A deposition nearby**. We investigated the hypothesis if different stop codons could affect the m$^6$A deposition. We categorized all the coding genes based on their stop codon (TAA/TAG/TGA), and investigated the m$^6$A probability value distribution in the region flanking the stop codon (Fig. 6a). We found three positions at which the m$^6$A sites could be adjacent to the stop codon. If TRR is the stop codon, then (1) Position −2 (Fig. 6a), straddled by motif NRACTRR; (2) position 1, straddled by TRACN; and (3) position 3, straddled by TRRACN. All the three positions showed relatively higher m$^6$A probability than other positions near stop codons (Fig. 6a for mouse, Supplementary Fig. 6a for human). Interestingly, transcripts with the TGA stop codon had higher m$^6$A probability as calculated by iM6A for all the three positions in comparison to transcripts with the TAA or TAG stop codon, particularly for the Position −2: NRACTRR and the Position 2: TRACN (Fig. 6a for mouse, Supplementary Fig. 6a for human). Next, we performed stop codon swaps and evaluated the resulting impact on m$^6$A deposition (Fig. 6b). We found that the m$^6$A probability at all the three locations decreased when the stop codon was changed to TAA or TAG, particularly for Position −2: NRACTRR and Position 2: TRACN (Fig. 6b, c for mouse, Supplementary Fig. 6b, c for human). Conversely, m$^6$A probability at the three locations increased when TAA or TAG was changed to TGA, again particularly for Position −2: NRACTRR and Position 2: TRACN (Fig. 6b, d for mouse, Supplementary Fig. 6b, d for human). Both stop codon swap experiments support that the TGA stop codon may favor m$^6$A deposition at and adjacent to a stop codon location. We further categorized the transcripts into

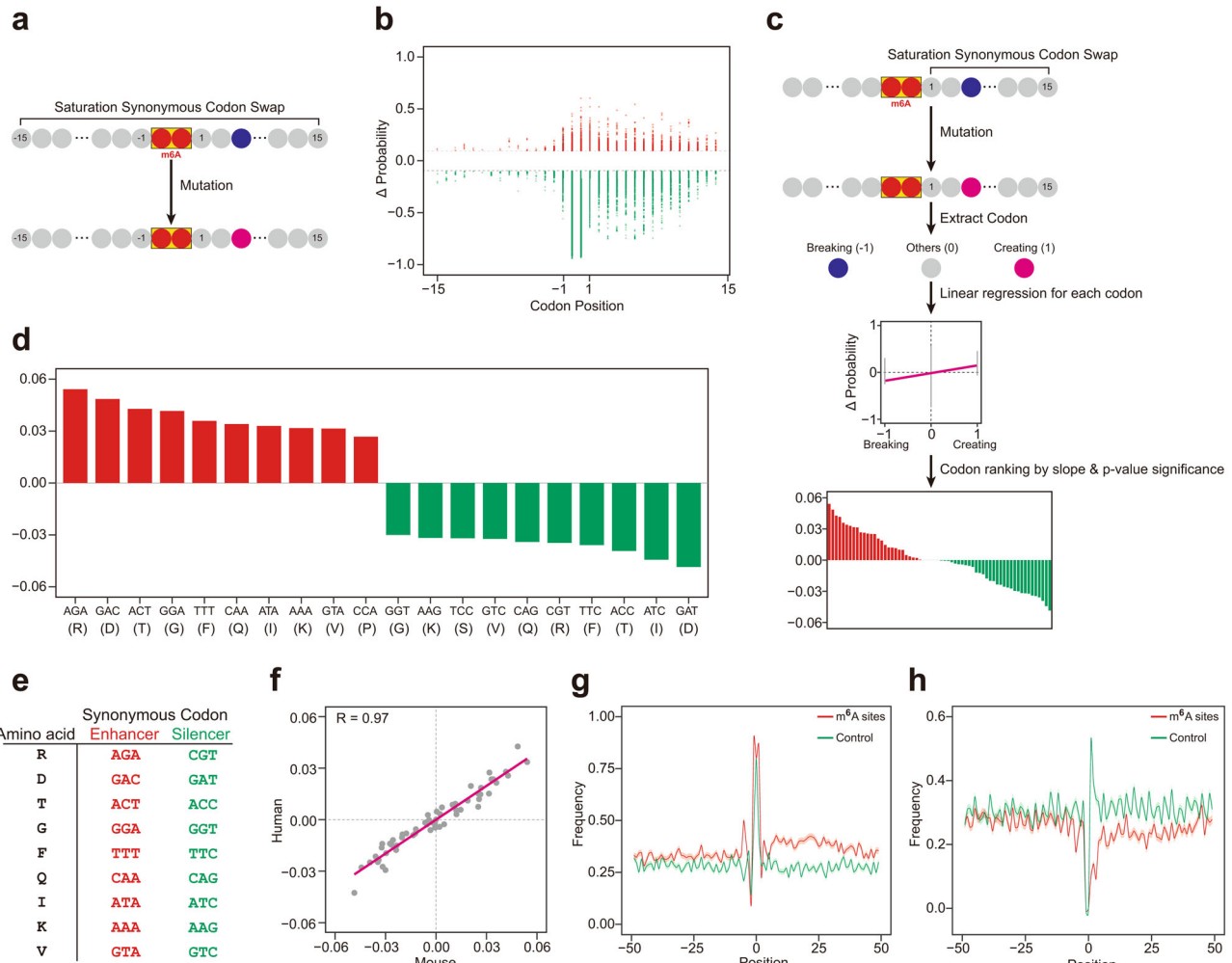

**Fig. 5 Synonymous codons may influence m⁶A deposition. a** The saturation synonymous codon swap strategy. For each codon in codons (−15 to +15 codons, −45 nt to +45 nt) around the m⁶A site, the codon was substituted by each of its synonymous codons. The delta changes of m⁶A probability value (ΔProbability) after swap was calculated by iM6A. **b** Positional plot of ΔProbability (cutoff = 0.1) for saturation synonymous codon swap. Red color dots were those codon swap events that increased m⁶A probability; Green color dots were those codon swap events that decreased m⁶A probability. **c** The systematic and quantitatively determination of the effect on m⁶A deposition for all codons by linear regression (See details in Methods). Total number of m⁶A sites used was 1473. (The full list of effect values for each synonymous codon are listed in Supplementary Data 3). **d** Bar plot of effect values for top 10 enhancer codons and top 10 silencer codons, its corresponding amino acids were also labeled. **e** Synonymous codon pair in enhancing and silencing m⁶A deposition. The amino acids (the first column) were coded by the synonymous codons (the second and third columns, respectively), which the enhancer codons enhancing m⁶A deposition, and the silencer codons silencing m⁶A deposition. **f** Scatter plot for the effect correlation for all synonymous codons between the study of human synonymous codons and the study of mouse synonymous codons. The effect of each synonymous codon was determined by the slope of its linear regression equation, and each gray dot was a synonymous codon. **g, h** Positional plot for the frequency of Top 20 enhancer codons (**g**) or silencer codons (**h**) in the sequences around the m⁶A sites. The plots were compared between higher m⁶A probability sites (red color, probability ≥ 0.7) and lower m⁶A probability sites (the exact RAC motif matched control, probability < 0.01). Data were presented as mean ± SEM. (Using other top number of enhancer or silencer codons generated similar results).

two groups according to their stop codons (TGA or non-TGA), and found that transcripts with the m⁶A sites at and adjacent to stop codons were statistically enriched with the TGA stop codon ($P < 1 \times 10^{-4}$, Fisher's exact test) (Fig. 6e for mouse, and Supplementary Fig. 6e for human). As evolution conservation provides evidence for functional importance, we further explored the conservation of stop codons in all the transcripts. Indeed, the stop codon of a transcript was more conserved if it was a part of an m⁶A site, supporting its functional importance (Fig. 6f and Supplementary Fig. 7a for mouse, Supplementary Figs. 6f, 7b for human). In the situation that the stop codon of a transcript was be part of an m⁶A site, the TGA stop codons were more conserved than the non-TGA stop codons, supporting that the TGA stop codon may favor the m⁶A deposition at and adjacent to stop

codon (Fig. 6f and Supplementary Fig. 7a for mouse, Supplementary Figs. 6f, 7b for human). Moreover, the TGA as a trimer motif may promote m⁶A deposition in comparison to TAA and TAG trimers (Supplementary Table 1).

**Evidence for an evolutionarily conserved m⁶A regulatory code in mouse and human.** For all the findings in this work, our data consistently suggests that the same m⁶A *cis*-element code governs m⁶A deposition in both human and mouse. To comprehensively address this hypothesis, we implemented a head-to-head test comparison for the human iM6A model and the mouse iM6A model, both of which were trained on that species' genes from most chromosomes except chromosome 9. Thus, the genes from human chromosome 9 and mouse chromosome 9 offered two

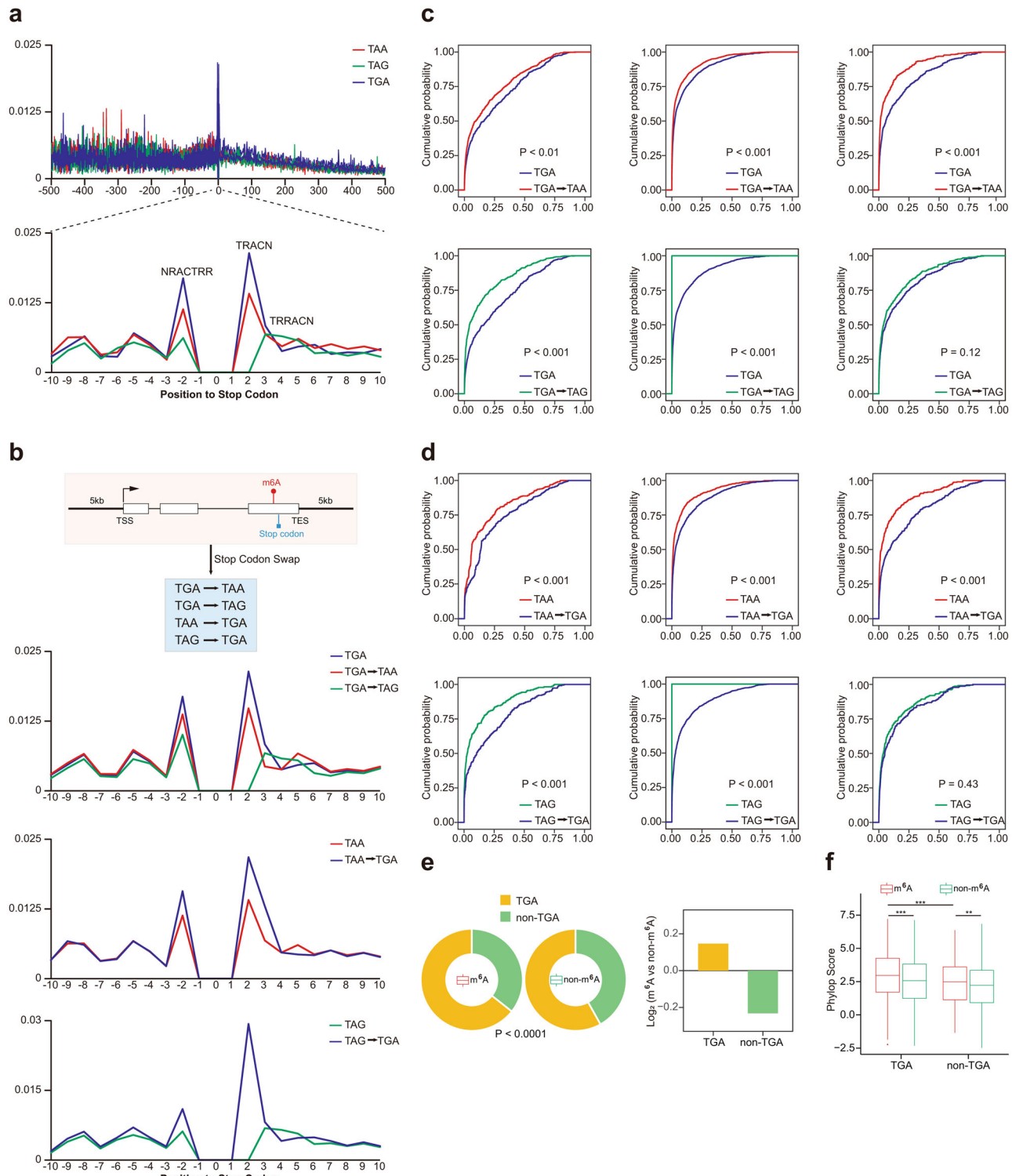

independent testing datasets being untouched for both the human and mouse iM6A models during their training. Human and mouse iM6A model performed comparably on both chromosome 9 transcripts (Fig. 7a, b), supporting that a common *cis*-element code governed the m6A deposition in both human and mouse. To further investigate the possibility of a common *cis*-element code, we compared the protein sequence conservation between human and mouse for the known components of the m6A methyl-transferase complex including METTL3, METTL14, WTAP, and VIRMA (Fig. 7c), and found that >95% of amino acids were

identical for each of the four proteins between human and mouse, supporting their functional conservation and, therefore, the likelihood of the *cis*-elements code commonality.

Our iM6A method modeled the m6A site-specific deposition in the pre-mRNA transcript, showing that the *cis*-elements regulating m6A deposition located preferentially within 50 nt downstream of m6A sites. It also identified which pentamers were m6A enhancers and silencers with the former mostly being part of RRACH motif and the latter mostly containing CG/GT/CT dinucleotides (Fig. 7d).

**Fig. 6 Stop codon TGA may favor m⁶A deposition at and adjacent to Stop codon. a** Positional plot of average modeled m⁶A probability around the stop codon, the position 0 was the T nucleotide for the stop codons. The red, green, and blue lines represented genes with TAA, TAG, or TGA as its stop codon respectively. Up panel: regions 500 nt upstream and downstream from 0 position. Bottom panel: regions 10 nt upstream and downstream from 0 position. **b** Positional plot for stop codon swap. First panel: illustrator of stop codon swap. Second panel: positional plot for TGA to TAA or TAG. Third panel: positional plot for TAA to TGA. Fourth panel: positional plot for TAG to TGA. **c** Cumulative distribution function (CDF) plot of modeled probability for TGA to TAA or TAG. The *p*-values were calculated by the Kolmogorov–Smirnov test (KS-test). (the left, middle, and right panel for NRAC<u>TRR</u>, <u>TRACN</u>, and <u>TRRACN</u> motifs). **d** Cumulative distribution function (CDF) plot of modeled probability for TAA or TAG to TGA. The *p*-values were calculated by the Kolmogorov–Smirnov test (KS-test). (the left, middle and right panel for NRAC<u>TRR</u>, <u>TRACN</u>, and <u>TRRACN</u> motifs). **e** The m⁶A sites were categorized into two groups (m⁶A or non-m⁶A) based on its probability value (the cutoff = 0.05), donut plot of percentage of stop codon for m⁶A sites and non-m⁶A sites (Left panel). Bar plot of log₂(odd ratio, m⁶A sites over non-m⁶A sites) of percentage of stop codon (Right panel). The *p*-value was calculated by the Fisher's exact test. **f** Box plot of conservation score of stop codons with or without m⁶A sites (*n* = 1311 for TGA with m⁶A sites, *n* = 1562 for TGA without m⁶A sites, *n* = 726 for non-TGA with m⁶A sites, *n* = 1125 for TGA without m⁶A sites). Median and interquartile ranges are presented for the box plot. The *p*-values were calculated by the one-sided Student's *t*-test (Significance: **\*\*P < 0.01, \*\*\*P < 0.001**).

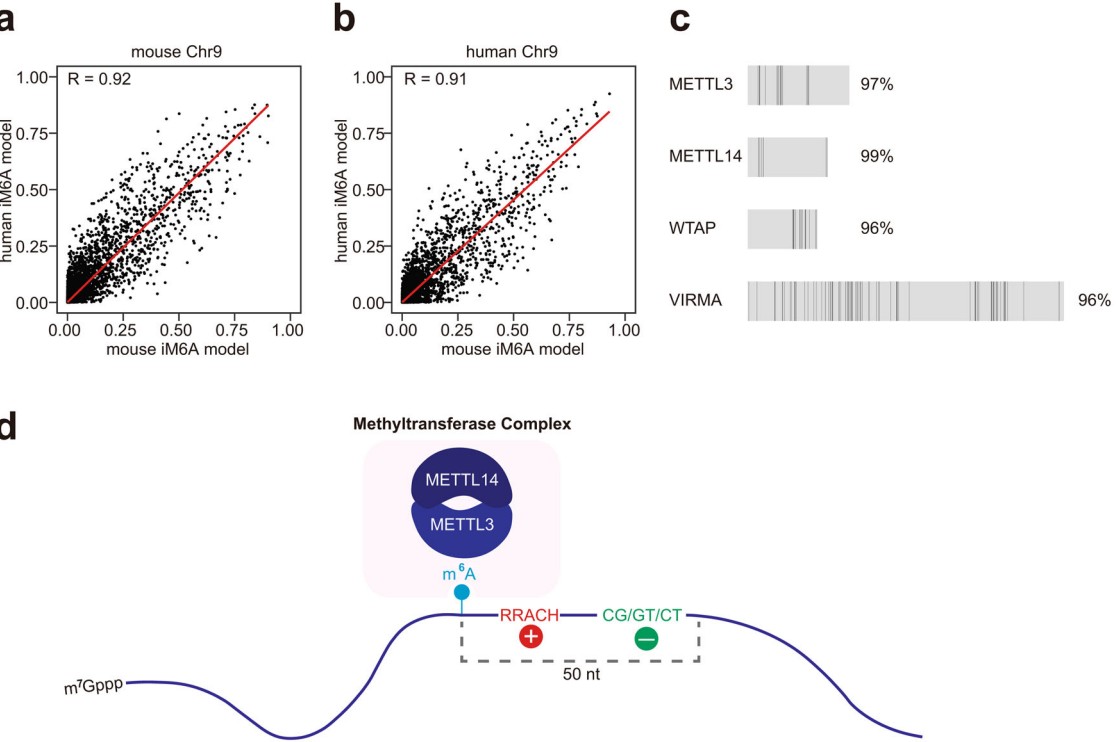

**Fig. 7 Evidence for an evolutionarily conserved m⁶A regulatory code in mouse and human. a**, **b** Scatter plot of modeled probability for the m⁶A sites (*n* = 100,000) in mouse chromosome 9 (**a**) or human chromosome 9 (**b**) using mouse iM6A model versus human iM6A model. Both mouse and human iM6A models were trained independently on data of other chromosomes except chromosome 9 in mouse and human genome respectively. Each dot represented one site in mouse chromosome 9 discovered by both models, and the labeled axes provided the probability values for that site by the two models. **c** The protein sequences conservation of METTL3, METTL14, WTAP, and VIRMA between human and mouse. The full length of the proteins was illustrated by the gray box with individual different amino acids marked as black lines. **d** *Cis*-elements that regulate m⁶A deposition locate majorly within 50 nt downstream of m⁶A sites. The m⁶A enhancers mostly contained part of the RRACH motif; the m⁶A silencers mostly contained the CG/GT/CT dinucleotides.

## Discussion

In this study, we used experimentally determined m⁶A sites from mouse and human as the training dataset to build our iM6A deep learning model, the modeling of which was confirmed to be accurate with AUROC = 0.99 by the independent testing dataset and by using m⁶A sites experimentally determined by a variety of precise m⁶A mapping methods. Taking advantage of the high modeling accuracy of the iM6A deep learning model, we implemented saturated mutagenesis on input transcripts to systematically perturb the iM6A deep learning black box. This led to discovering that the downstream 50 nt of the m⁶A sites located a high density of *cis*-elements regulating m⁶A deposition. Applying the linear regression model as a machine learning method to the saturated mutagenesis data, we were able to systematically

identify m⁶A enhancers and silencers in the region. Thorough bioinformatics characterization of these *cis*-elements including positional plot and sequence conservation analysis confirmed their designated function. Our finding was further supported by independent experimental validations, and uncovered a key *cis*-regulatory mechanism for m⁶A site-specific deposition.

Though deep learning modeling is powerful in integrating large datasets and accurate in modeling compared to traditional machine learning methods, it is hard to interpret the underlying biological insights from its deep learning model as a black box. Conversely, traditional machine learning approaches such as linear regression are useful for connecting model parameters to the biological function. In this study, we took advantage of both deep learning and machine learning: we first implemented the

iM6A deep learning model to accurately model m6A deposition, then applied linear regression as a machine learning approach to systematically characterize the *cis*-element contribution in the high-throughput saturated mutagenesis data from the iM6A deep learning modeling. Our joint method of combining both deep learning and machine learning may be applicable to other biological investigations.

Our work revealed that m6A *cis*-elements are largely located within the 50 nt region downstream of the m6A site. This 50 nt range echoes that of several other RNA processing events. A typical exon length is usually at least 50 nt or longer though tiny exons do exist[55]. The *cis*-elements that regulates cleavage and polyadenylation generally locate within 50 nt of the cleavage site[56]. The size 50 nt may reflect the physical sizes of the different RNA processing complexes. A detailed investigation of the molecular mechanism underlying the 50 nt range would be an interesting and useful future research project based on thorough biochemistry experiments. Also, our work revealed that the *cis*-elements regulating the m6A deposition include the RRACH enhancers and the CG/GT/CT silencers. What are the *trans*-factors that recognize these *cis*-elements and how they regulate m6A deposition are all potentially worth directions for future molecular mechanism investigations.

The iM6A modeling introduced here should prove effective in future mechanism investigations of m6A regulation and deposition because it not only enables accurate modeling of the m6A deposition but also offers a high-throughput, fast and efficient mechanism discovery which would be cost-prohibitive and time-impractical for traditional laboratory experimentation. Anticipating broad interests in our iM6A strategy, we have deposited our iM6A source code at GitHub (https://github.com/ke-laboratory/iM6A) as well as the probability values of m6A candidates in human and mouse genes (https://doi.org/10.5281/zenodo.4734266).

## Methods

### iM6A Model

*Model architecture.* The iM6A is based on a deep residual neural network[44]. The basic unit of iM6A is the Residual Network (ResNet) block and we implemented the ResNet structure according to the CNN Architectures and implementations by MLT (https://github.com/Machine-Learning-Tokyo/CNN-Architectures), which is composed of batch-normalization (BatchNorm) layers, rectified linear units (Relu), and convolutional (Conv1D) layers organized in a specific manner (Fig. 1a). In ResNet block, $k$, $w$, ar, and $r$ are the number of convolutional kernels, window size, dilation rate of each convolutional kernel in the layer, and repetition numbers, respectively. The current combination of $k$, $w$, ar, and $r$-values are showed in Fig. 1a, which were obtained by hyperparameter search (Supplementary Data 1). iM6A starts with a Conv1D, then is followed by four ResNet blocks. The output of every ResNet block is added to the input of penultimate layer (Cropping1D), connected with a Cov1D layer with softmax activation.

The training input of iM6A for each gene is the full length of the pre-mRNA sequence with 5000 nucleotides on each side, covering the transcript from 5 kb upstream of TSS (transcription start site) to 5 kb downstream of TES (transcription end site).The sequence is transformed by One-Hot-Encoding, which N, A, C, G, and T are encoded as [0,0,0,0], [1,0,0,0], [0,1,0,0], [0,0,1,0], and [0,0,0,1] respectively. Then, the one-hot-encoded nucleotide sequence was split into blocks of length 5000 + 5000 + 5000 in such a way that the $i^{th}$ block is consisted of the nucleotide sequence position from $5000(i-2) +1$ to $5000i + 5000$. Along with the sequence input, the location for the positive training set of individual m6A sites was marked out, and the output label was also split into block of length 5000 in such a way that $i^{th}$ block consists of the positions from $5000(i-1) +1$ to $5000i$. Information of input and output was jointly feed into the ResNet deep learning network for training. A similar input strategy has been implemented in SpliceAI[45] that implemented deep learning to model splice sites in pre-mRNA. The output of the model is the probability value of each position being an m6A site.

*Model training and testing.* We downloaded the gene annotation tables (vM7 for mouse, v19 for human) from GENCODE (https://www.gencodegenes.org/) and extracted the longest transcript for each gene. Both mouse and human m6A sites were collected from published data[8,14,32,33], which were determined by m6A-CLIP experiments. The consensus motif for an m6A site could be either RRACH as a high stringent set or RAC as a more inclusive set. We generated two separate iM6A

models using either stringency: the RRACH dataset for RRACH iM6A model and the RAC dataset for RAC iM6A model. The transcripts with its m6A sites were kept as input (mouse RRACH: 8475 genes, 41,551 m6A sites; mouse RAC: 8939 genes, 57,712 m6A sites; human RRACH: 8598 genes, 54,354 m6A sites; human RAC: 10,314 genes, 81,519 m6A sites). We used pre-mRNA sequences as input: the m6A sites on pre-mRNA were served as positive sites, while the remained nucleotides were treated as negative sites. The whole dataset was divided into training and test datasets. The training dataset contained all the transcripts on most chromosomes except chromosome 9, the transcripts of which were held out and reserved for the test later on.

The iM6A were trained for 10 epochs with a batch size of 30 on NVIDIA GPU. By pulling singularity container (tensorflow-19.01-py2) from NVIDA official website, we created an environment for model training and testing. Extra packages (biopython: 1.76; scikit-learn: 0.20.3, matplotlib: 2.2.4, keras: 2.0.5) were installed into an external path by pip. For training, Adam optimizer was used to minimize the categorical cross-entropy loss between the target and modeled outputs. The learning rate of the optimizer was set as 0.001 for the first 6 epochs, and then reduced by a factor of 2 in every subsequent epoch. We trained the model for five times and obtained five trained models. For testing, each input was evaluated using all five trained models, while the average score of their outputs was used as the modeled value.

*Comparison of iM6A with other methods.* We compared the modeling performance of iM6A with that of the machine learning-based SVM method[39] and that of the deep learning-based CNN-RNN method[40]. Both SVM and CNN-RNN models were trained on the same training samples used for iM6A, and the m6A and non-m6A sites were conformed to the RRACH motif in the same way as in Chen et al., 2019. For the positive training data, the input is the sequence centered on the m6A sites (39,138 sites). For the negative training data, the input is the sequence centered on the non-m6A sites, which were randomly selected from the non-m6A sites on the same full transcripts that contained the positive sites. The sequence length for SVM model was 41 nt as described in Chen et al., 2019, while the sequence length for CNN-RNN model was 1001 nt as in Wang and Wang, 2020. Moreover, the positive-to-negative ratio was 1:1. For independent testing, the sequence centered on the m6A and non-m6A sites in chromosome 9 were used to quantify the modeling performance of the models, and ROC (receiver operator curves) curves and AUROC (area under receiver operator curves) scores were used as the performance evaluation metrics.

*Validation of iM6A modeling by experimentally detected m6A sites.* We downloaded the gene annotation tables (vM7 for mouse, v19 for human) from GENCODE (https://www.gencodegenes.org/) and extracted the longest transcript for each coding gene (mouse: 22,357 genes, human: 20,536 genes). The probability value of each nucleotide being an m6A site in the pre-mRNA of the transcripts were modeled by iM6A, and the sites selected were those conforming to the RRACH (the iM6A RRACH model) or the RAC (the iM6A RAC model). We collected the m6A sites detected by the experimental methods including m6A-CLIP[8,14,32,33], m6a-label-seq[36], MAZTER-seq[34], and m6ACE-seq[35]. The heatmap was used to visualize the experimentally detected sites in all modeled sites. The modeled sites were ranked by its probability value, and the black line denoted whether methylation was identified by the experimental method at the site (Fig. 1c, d).

*Calculation of the m6A probability and the enrichment score for the m6A sites derived from m6A-CLIP.* The peak enrichment value for the m6A sites in mouse (mouse embryonic stem cell, mESC) and human (the A549 cell line) were quantified by the m6A-CLIP[14,33]. Based on the enrichment score, the m6A sites were categorized into three groups (low: score < 5, medium: 5 ≤ score < 20, high: score ≥ 20). The probability of the site being an m6A site was modeled by the iM6A, and the box plot was used to visualize the peak enrichment value and the modeled m6A probability (Fig. 1e and Supplementary Fig. 1h).

*Calculation of the m6A probability and the cleavage efficiencies for the m6A sites derived from MAZTER-seq.* The m6A sites identified by MAZTER-seq[34] were downloaded. According to their supplemental tables, the m6A sites were categorized into the groups of control, low, intermediate, high, and highest confidence. We filtered the dataset to retain the sites conforming to the RRACA motif and extracted the normalized cleavage efficiency of the sites from the table. Box plot was used to visualize the normalized cleavage efficiency and the modeled m6A probability (Supplementary Fig. 1i, j).

*Comparison of the RRACH model with the RAC model.* Both RRACH and RAC of the iM6A models for mouse (Fig. 1f) were trained independently on the genes of all the other chromosomes except those of the chromosome 9 (Chr9). The m6A sites in Chr9 were modeled by either the RRACH iM6A model or the RAC iM6A model, and the scatter plot was used to visualize the modeled probability of the m6A sites between the RRACH model and the RAC model. Each dot represented one site in Chr9 discovered by both models, and the labeled axes provided the probability estimate for that site by the two models. The R-value was calculated by Pearson Correlation Coefficient. The same analysis was performed for human (Supplementary Fig. 1k).

*Positional mutational effects on m6A deposition implemented by single nucleotide saturation mutagenesis.* For the m6A sites in last exon, we modeled its probability by iM6A. The sites were sorted based on probability value, and a single m6A site with the highest probability value were kept for each gene. In addition, the probability should be ≥0.4. Then, we selected the sites which located at least 250 nt away from both last exon start and last exon end. We obtained 2048 sites for mouse and 2724 sites for human in the last exon region. The same strategy was applied to the m6A sites in long internal exon, and we obtained 893 sites for mouse (the m6A sites in *Plekhm3* gene was excluded for its unusual sequence property) and 806 sites for human.

For each position in the sequence (−250 to 250) flanking the m6A site, the nucleotide was substituted by each of the three other nucleotides (Fig. 2a). The delta changes of m6A probability value (ΔProbability) after mutation was calculated by iM6A (Fig. 2a).

*Quantify the effect of all cis-element pentamers by linear regression.* To prepare the m6A sites in last exon for the systematic effect analysis of all pentamers, we first modeled the m6A probability by iM6A for all m6A sites in last exons. In addition, the probability should be ≥0.4. All of these sites were sorted based on their m6A probability value, and only a single m6A site with the highest probability value was kept for each gene. We further selected the m6A sites which located at least 50 nt away from both last exon start and last exon end. We obtained 5292 sites for mouse and 4772 sites for human from which we randomly selected 1500 sites for both mouse and human, as 1500 sites was sufficient for our analysis. The same strategy is applied to the m6A sites in long internal exon (length > 100 nt), and we got 1460 sites for mouse and 1416 sites for human.

For each position in the downstream region of an m6A site (i.e., from position 3 to position 46), the nucleotide was substituted by each other nucleotides. The resulted probability change (ΔProbability) of this m6A site is calculated by iM6A. Each substitution created and broke 5 overlapping 5-mers simultaneously, and −1 or 1 was assigned to each of the five created or broke 5-mers. Linear regression was implemented to each 5-mers (total 1024 pentamers) when pooling all the data, then the effect of each motif was ranked based on the slope of linear regression equation and the statistical significance was quantified by *p*-value (Fig. 2d).

*Positional plot of pentamers in sequences flanking m6A sites.* For the potential m6A sites in the RAC consensus at the last exons of each gene, we calculated their m6A probability values by iM6A. The m6A sites were sorted based on their m6A probability value, and a single m6A site with the m6A highest probability value were kept for each gene. We selected the m6A sites with the higher m6A probability values (probability ≥ 0.7) as the positive sites, while the control was the exact RAC motif matched site with a lower m6A probability value (probability < 0.1). For the m6A enhancer and silencer positional plot, we randomly selected 1000 positive sites or control sites located in the last exon, and extracted the 50 nt upstream and downstream sequence flanking the m6A site. The pentamers were enumerated from the 5' end to the 3' end of the 101 nt sequence. For the positional plot, we counted the numbers of top 100 enhancers and top 100 silencers at each position of the 101 nt sequence (see details in the section for quantifying the effect of all *cis*-element pentamers). The frequency of the enhancers or silencers were also calculated. The plots were compared between the positive sites and the control, and the data were presented as mean ± S.E.M. standard error of the mean (Fig. 2h, i).

In parallel, we collected the m6A sites detected by different experimental methods, including m6A-CLIP[8,14,32,33], m6a-label-seq[36], m6ACE-seq[35], and MAZTER-seq[34]. The experimentally determined m6A sites (m6A-CLIP, m6A-label-seq, m6ACE-seq, and MAZTER-seq) served as the positive sites while the control was the exact RAC motif matched site which was not determined by the experimental methods. Moreover, those control sites did not overlap with the m6A peak regions[14,33] and came from the transcripts that also contained the positive sites. For the sites detected by MAZTER-seq, we intersected it with the sites determined by other methods (m6A-label-seq, m6ACE-seq, and m6A-CLIP) to get the high-quality sites as to lower the multiple technical noises of MAZTER-seq as discussed in Garcia-Campos et al. 2019[34]. The overlapped sites were served as the positive sites while the control was the exact RAC motif matched site which was not determined by the MAZTER-seq. In addition, those control sites did not overlap with the m6A peak regions[14,33] and came from the transcripts that also contained the positive sites. For the m6A enhancer and silencer positional plot, we selected the positive sites or control sites located in the last exon, and extracted the 50 nt upstream and downstream sequence flanking the m6A site. The pentamers were enumerated from the 5' end to the 3' end of the 101 nt sequence. For the positional plot, we counted the numbers of top 100 enhancers and top 100 silencers at each position of the 101 nt sequence (see details in the section for quantifying the effect of all *cis*-element pentamers). The frequency of the enhancers or silencers were also calculated. The plots were compared between the positive sites and the control, and the data were presented as mean ± S.E.M. (Supplementary Fig. 3).

*Conservation analysis for the sequence flanking the m6A sites.* For the potential m6A sites in the RAC consensus at the last exon, we calculated their m6A probability values by iM6A. The m6A sites were sorted based on their m6A probability value, and a single m6A site with the m6A highest probability value were kept for each

gene. We selected the m6A sites with the higher m6A probability values (probability ≥ 0.7) as the positive sites, while the control was the exact RAC motif matched site with a lower m6A probability value (probability < 0.1). In addition, these RAC sites were located in the noncoding region of last exon (at least 50 nt from the stop codon). We calculated the phyloP score of each nucleotide flanking the RAC sites. The average phyloP score for the sequence flanking the RAC sites were calculated, the plots were compared between the positive sites and the control, and the data were presented as mean ± S.E.M. (Fig. 2j).

*Experimental validation of iM6A modeling by the m6A profiling in the lymphoblastoid cell lines (LCLs) of 60 Yoruba (YRI) individuals.* The m6A levels were profiled across the transcriptome in LCLs derived from 60 YRI individuals[50]. We downloaded raw sequencing data from Gene Expression Omnibus (GEO) repository (GSE125377). Raw sequencing data was mapped to the hg19 reference genome by HISAT2 with the parameter "-known-splicesite-infile <splice-file extract from Refseq hg.19 GTF file > –k 1 —no-unal". The BAM files obtained from the alignment were used as an input file for BigWig file, which were visualized by UCSC Genome Browser (Fig. 3).

The m6A profiling dataset in the LCLs of 60 YRI individuals was downloaded from Zenodo (https://doi.org/10.5281/zenodo.3870952), which includes the bed file of m6A peaks, the normalized enchainment score of each peak of 60 samples, and the imputed genotype data of 60 samples. We downloaded all SNP sites from dbSNP database (https://ftp.ncbi.nih.gov/snp/organisms/human_9606_b151_GRCh37p13/BED/), and extracted all the SNVs located in the m6A peaks. The corresponding genotype of each SNV for 60 samples were also extracted. For each SNV, 0, 0.5, and 1 were assigned to homozygote of the major allele, heterozygote, and homozygote of the minor allele. The association between SNV and m6A level was tested by linear regression. We obtained 3297 SNVs that were strongly correlated with m6A level (*p*-value ≤ 0.1). Then, we calculated the effects of these SNVs on m6A deposition by iM6A, and found 47 SNVs which could affect m6A deposition significantly (|ΔProbability| ≥ 0.1). The delta changes of peak enrichment (ΔPeakEnrichment) of these 47 SNVs corresponding m6A peaks were calculated using the m6A profiling experimental data (Fig. 3).

*Characterization of m6A associated SNVs.* We downloaded data from ClinVar database (https://ftp.ncbi.nlm.nih.gov/pub/clinvar/tab_delimited/variant_summary.txt.gz), and extracted the SNVs located in last exons. We obtained 68286 SNVs, and utilized iM6A to calculate their effects on m6A deposition. To characterize the SNVs that altered m6A deposition, we selected the SNVs located within 500 nt upstream or downstream of an m6A site. Then those SNVs were categorized by clinical significance according to ClinVar, and we only kept the sites annotated with uncertain significance/benign/likely benign/pathogenic/likely pathogenic. We singled out uncertain significance as the first group (VUS), grouped benign and likely benign as the second group (Benign), and grouped pathogenic and likely pathogenic as the third group (Patho.)(Fig. 4). Then, the SNVs were categorized into two groups (m6A probability changed group or no change group) based on ΔProbability (|ΔProbability| ≥ 0.1 for the changed group). Bar plot was used to show the $\log_2$(odd ratio, m6A probability changed group over no change group) for the percentages of SNVs with different clinical significances in ClinVar, and the *p*-value was calculated by the Fisher's exact test. To visualize the effect of SNVs on m6A deposition, saturation mutagenesis was performed in the region −100 to +100 nt up- and downstream of m6A sites, and the ΔProbability of each mutation event was displayed as a heatmap (Fig. 4).

*Saturation synonymous codon swap.* For the m6A sites in last exons, we calculated m6A probability values by iM6A. Then the m6A sites were sorted based on their probability value, and a single m6A site with the highest probability value was kept for each gene. Then, we selected the m6A sites which located at least 60 nt away from both the last exon start and the end of the coding sequence. In total, we obtained 1473 m6A sites for mouse and 1532 m6A sites for human in the last exon region. For each codon position in the codons (−15 to +15 codons, −45 nt to +45 nt) flanking the m6A site, the codon was substituted by each of its synonymous codons (Fig. 5a). The delta changes of the m6A probability value (ΔProbability) after the codon swap was calculated by iM6A.

*Quantify the effect of all synonymous codons on m6A deposition by linear regression.* To prepare the m6A sites in last exons for the systematic effect analysis of all pentamers, we first modeled the m6A probability value by iM6A for all m6A sites in last exons. These sites were sorted based on their m6A probability value, and only a single m6A site with the highest probability value was kept for each gene. We further selected the m6A sites which located at least 60 nt away from both the last exon start and the coding sequence end. In total, we obtained 1473 sites for mouse and 1532 sites for human. For each codon position in the downstream region of an m6A site (from position 1 to position 15), the codon was substituted by each of its synonymous codons. The resulted probability change (ΔProbability) of this m6A site was calculated by iM6A. Each codon substitution created one codon and simultaneously broke the original codon. A value of 1 or −1 was assigned to the created codon or the broken codon accordingly. Linear regression was implemented to each codon (total 64 codons) when pooling all the data together, then the effect of each synonymous codon was ranked based on the slope of the linear

regression equation and the statistical significance was quantified by *p*-value (Fig. 5c).

*Positional plot of trimers in sequences flanking m6A sites*. For the potential m6A sites in the RAC consensus at the coding region of the last exon for each gene, we modeled its probability value by iM6A. We selected the m6A sites with relatively high probability values (probability ≥ 0.7) as the positive sites, while the control was the exact RAC motif matched site with lower m6A probability (probability < 0.01). For the trimers enumeration, we extracted the 50 nt upstream and downstream sequences of each m6A site. The trimers were enumerated from the 5' end to the 3' end of the 101 nt sequence. For the positional plot, we counted the numbers of the top 20 enhancers codons and the top 20 silencers codons at each position of the 101 nt sequence (see details in the section for quantifying the effect of all synonymous codons on the m6A deposition by linear regression). The frequency of enhancers or silencers were also calculated. The plots were compared between the positive sites and control, and the data were presented as mean ± S.E.M. (Fig. 5g, h and Supplementary Fig. 5c, d).

*Distribution of m6A probability value around stop codon*. All the coding genes were categorized by their stop codons (three groups: TAA/TAG/TGA). To plot the distribution of m6A probability value around stop codon (the position 0 was the T nucleotide for stop codons), we first calculated the m6A probability value of each nucleotide flanking the stop codon (−500 nt to +500 nt) by iM6A. The total probability value for each position was summed, and the average probability value was computed by dividing the total number of transcripts at each position (Fig. 6a).

For the stop codon swap, the TGA stop codon was substituted by the TAA or the TAG stop codon (Fig. 6b). Similarly, TAA or TAG was replaced by TGA. The average m6A probability value around the stop codon was also calculated by the iM6A. For the m6A sites adjacent to the stop codon (Position −2: NRACTRR, Position 2: TRACN, Position 3: TRRACN), we evaluated the m6A probability value changes by the CDF (Cumulative Distribution Function) plot, and the *p*-values were calculated by the KS-test (Kolmogorov-Smirnov test).

*Conservation analysis of stop codons with or without m6A sites*. For the RAC sites adjacent to the stop codons (NRACTRR, TRACN, and TRRACN, and TRR represented the stop codon), we calculated the average phyloP score of its corresponding stop codon. The m6A sites were categorized into two groups (m6A or non-m6A) based on its probability value (the cutoff = 0.05), while its corresponded stop codon was also categorized into two groups (TGA or non-TGA). The donut plot was used to show the percentage of stop codon for the m6A sites and the non-m6A sites, and the *p*-value was calculated by the Fisher's exact test. The conservation score of stop codons for each group was compared by the box plot, and the *p*-value was determined by the Student's *t*-test.

*Conservation analysis of stop codons with or without m6A-CLIP sites*. For the RAC sites adjacent to or at the stop codons (NRACTRR, TRACN, and TRRACN, and TRR represented the stop codon), we calculated the average phyloP score of its corresponding stop codon. The stop codons adjacent to or with the m6A-CLIP sites were the stop codons that overlapped with either the m6A sites detected by m6A-CLIP at the three positions or the m6A peak region[8,14,32,33]. The control was the stop codons that was neither adjacent to nor at the m6A-CLIP sites nor overlapped with the m6A peak regions[14,33], and we further require the control stop codons to come from the transcripts with the m6A-CLIP sites (i.e., these transcripts had adequate expression level to have m6A sites detected by the m6A-CLIP). Moreover, the stop codons were categorized into two groups (TGA or non-TGA). The conservation score of stop codons for each group was compared by the box plot, and the *p*-value was determined by the Student's *t*-test.

*Comparison of mouse iM6A model with human iM6A model*. Both mouse and human iM6A models were trained independently on the genes of all the other chromosomes except the chromosome 9 (Chr9) in mouse and human, respectively. The m6A sites in Chr9 of mouse or human were modeled by either mouse iM6A model or human iM6A model independently, and the scatter plot was used to visualize the modeled probability of the m6A sites between the mouse and human models (Fig. 7a, b). Each dot represented one site in chromosome 9 (Chr9) discovered by both models, and the labeled axes provided the probability estimate for that site by the two models. The *R*-value was calculated by Pearson Correlation Coefficient.

*Comparison of the protein sequence conservation of METTL3, METTL14, WTAP, and VIRMA between mouse and human*. The mouse and human protein sequence of METTL3 (Mouse: Q8C3P7, Human: Q86U44), METTL14 (Mouse: Q3UIK4, Human: Q9HCE5), WTAP (Mouse: Q9ER69, Human: Q15007), and VIRMA (Mouse: A2AIV2, Human: Q69YN4) were downloaded from Uniport (https://www.uniprot.org/). The MEGA (Molecular Evolutionary Genetics Analysis) software[57] was used to align the protein sequences. We visualized the sequence conservation with the heatmap. The full length of the protein was illustrated by the gray box, while the individual amino acids that differed between mouse and human

were marked as black lines. The percentage of conserved amino acids between mouse and human proteins was also calculated accordingly (Fig. 7c).

**Reporting summary**. Further information on research design is available in the Nature Research Reporting Summary linked to this article.

## Data availability

The datasets of the probability for the m6A candidates are deposited to available at Zenodo (https://doi.org/10.5281/zenodo.4734266). ClinVar dataset is available at https://ftp.ncbi.nlm.nih.gov/pub/clinvar/tab_delimited/variant_summary.txt.gz. dbSNP dataset is available at https://ftp.ncbi.nih.gov/snp/organisms/human_9606_b151_GRCh37p13/BED/. The m6A profiles of 60 YRI samples were available with the Gene Expression Omnibus repository under accession no. GSE125377 and https://doi.org/10.5281/zenodo.3870952.

## Code availability

The source code of iM6A is available at GitHub (https://github.com/ke-laboratory/iM6A).

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

## Acknowledgements

We thank Liangjiang Wang for sharing the code of TDm6A, Jia Meng for sharing the code of WHISTLE. We thank Dennis Weiss, Susan E. Liao, and members of Ke Laboratory for comments, suggestions, and thoughtful discussions. Ke Laboratory and this research is funded by NIH/NIGMS Maximizing Investigators' Research Award (MIRA) R35 Award (R35 GM133711 to S.K.), American Cancer Society Pilot Award (ACS-2019-Pilot-Ke/IRG-16-191-33/ IRG-21-136-36-IRG to S.K.) and the Jackson Laboratory Cancer Center New Investigator award from the NIH/NCI Cancer Center Support Grant (2 P30 CA034196-34 to S.K.).

## Author contributions

S.K. and Z.L. conceived and designed the study and wrote the manuscript. Z.L. conducted the experiments and performed the data analysis, with some additional contribution from J.Z. and J.F. in experimental validation design and test. S.K. supervised the research.

## Competing interests

The authors declare no competing interests.
