## [Peer Review File · Nature Communications]

Title: Deep learning modeling m6A deposition reveals the importance of downstream cis-element sequencesREVIEWER COMMENTS

Reviewer #1 (Remarks to the Author):

This paper by Luo and Ke presents a new approach to sequence-based m6A site prediction and analysis with helpful comparisons between species and to sites identified using various sequencing approaches. However, as noted in the introduction, other m6A site predictors have been developed. At this point, the locations of m6A have been well-characterized (or at least often characterized) in both human and mouse transcriptomes. Of note this year, Körtel et al. (2021) developed an updated protocol for miCLIP and used this data in combination with a machine learning approach (also with an AUC of 0.99) to identify m6A sites in four human and mouse cell lines. It is unclear how iM6A compares, and biological insights from sequence modifications seem to primarily capture the known importance of RRACH motifs.

Major questions/comments:

1. Do negative training sites centre on adenosine residues and RAC or RRACH motifs?
2. If using 5000 nucleotides on either side of pre-mRNA as training input, how are sites encoded that don't have 5000 bases on both sides? Is this information used by the model and could it bias the models towards the prediction of m6A sites in the last exon?
3. Körtel et al. claim that their method of filtering training data for positive and negative sites offers improvements over standard m6A-CLIP/miCLIP analysis methods. Does using their sites to train iM6A change predictions (i.e. how sensitive is the method to differences in training data)?
4. Based on their miCLIP2 data, Körtel et al. predict many non-DRACH motif sites as methylated. What percentage of these are captured by the iM6A model? Does iM6A predict any non-RAC motifs as methylated?
5. Fig 1d: in comparison to MAZTER-seq and other methods, rather than considering the rank of sites, what percent of sites experimentally detected as methylated were predicted to be methylated by iM6A? Can you show a scatter plot of rank vs. probability for these sites (and a distribution of probabilities for tested sites overall)? How do probabilities from iM6A compare to methylation rates per site quantified using MAZTER-seq?
6. Fig 2: Figure 2h suggests that the top enhancer motifs frequently encompass the methylated residue. As RRACH motifs are already widely recognized to be important for methylation, how does excluding sites directly surrounding the methylated adenosine change the "enhancer" and "silencer" sets? If RRACH motifs are still among the top enhancers, would this be because they are themselves methylated? Or could this be an artifact of multiple m6A sites in close proximity increasing peak enrichment in miCLIP data without necessarily affecting one another's methylation rates?
7. (Lines 345-346) Do the SNVs discussed as putative factors in disease through changes in m6A lead to synonymous mutations? Can effects from changes at the epitranscriptomic level be separated from sequence changes at the protein level? In this section of the text in general, do any of the SNVs detected lead to synonymous mutations?
8. In Fig 3f/h & 4e/g, examples focus on direct changes to the RRACH motif encompassing methylated sites. It is not surprising that these would affect methylation rates. The same is true in Figure 6, focusing on stop codons. Again, stop codons seem to be important mainly

as part of RRACH motifs or their direct extensions, with differences falling off within 3 nucleotides of methylated sites (Figure 6a). Do conclusions regarding stop codon associations with m6A represent anything beyond the likelihood of various stop codons to be included in or to disrupt favourable RRACH motifs?

(Lines 427 & 430): it seems particularly obvious that a change in the stop codon at position 2 that changes A to G would abrogate m6A methylation at that A (there is no longer an adenosine to be methylated), while a change from G to A would have the opposite effect.

9. Regarding Figure 7, could errors when applying cross-species model reveal anything about species-specific m6A deposition preferences?

10. If it is true that sequence changes downstream of m6A sites have a greater effect on m6A probability than sequence changes upstream, any predictions as to mechanism?

11. Given that many previously published m6A site predictors also claim high accuracy, could the same biological insights have been generated based on these tools rather than iM6A? What is the overlap in sites predicted using these different methods?

12. m6A methylation rates are reported to differ across cell lines (e.g. Liu et al., 2013 SCARLET paper). Does accuracy decrease if iM6A is trained on data from one cell line and tested on data from another?

Minor comments:

- Table S2: Is the p-value really exactly 0 for all pentamers?
- Ext Fig 3: a lot of text in the legend is repeated for each subfigure and could be condensed.

Reviewer #2 (Remarks to the Author):

The manuscript presented a deep learning model for m6A site prediction and interpretation. In general, the manuscript is well written, the experiments were well designed with multiple different technical considerations.

The modeling of m6A sites using whole Pre-mRNA is indeed novel and can potentially bring major improvement in performance. Many experiments were designed to show the performance of the predictor and great efforts were made to interpret the model and its biological meaning. I did see great efforts of the authors and truly appreciate that.

Nevertheless, I have the following comments.

1. Truly sorry for being skeptical. The reported performance (AUROC=0.99) seems too good to be true based on existing literature [1]. This is due to the highly imbalanced dataset, with a lot more negative sites than positive sites. AUROC is known to be strongly affected by data imbalance. It is better to calculate the AUPRC, which is not effected by sample imbalance, and also report the AUROC on balanced data (with the same number positive and negative sites or samples). Ideally, the same number of negative and positive sites should be extracted from the same transcript as in most existing works for RNA modification site prediction.

2. “pre-mRNA” has been used to describe the input sequence. Do you also consider lncRNA? Can the negative sites be from intronic regions, which are often easier to recognize and leads to higher reported performance?
3. Figure 1b is not accurate. WHISTLE achieved good performance because it also takes advantages of genomic features. If you use only SVM and sequence information, then it is just MethyRNA method.
4. Authors showed a potential function of iM6A to identify the m6A associated SNVs (Figure 4). There have already been quite a few works in this field, which should probably be mentioned, including at least [2, 3].
5. Please provide more details related to the encoding of m6A sites. In the METHOD section. Authors described that ‘The training input of iM6A for each gene is the full length of the pre-mRNA sequence with 5000 nucleotides on each side. This means the input are not of the same dimension for different genes. How do you deal with super long transcripts (100 thousands nt)? Do you consider pre-mRNA only, or you also considered pre-lncRNA. How do you label the m6A sites in the encoded information (it should be part of the input, and not clear to me based on Figure 1a or Figure 6b)?
6. Please provide your training and testing data as well, so that people can fully replicate the reported performance. (Apologize for being skeptical. But I am sure that you don’t want to find out later that the high performance is due to unfair setting or a mistake.)
7. If the cis-elements regulating the m6A deposition preferentially reside within the 50 nt downstream of the m6A sites, why the input sequence of your model is so long (at least 10kb)? Most existing method uses only 1kb sequences. Could you test the performance of your method with shorter sequences on both sides?
8. Sorry for my ignorance. How do you define “m6A enhancers” and “m6A silencers”? This is the first time I saw these two terms.

Reference

1. Chen, Z., et al., Comprehensive review and assessment of computational methods for predicting RNA post-transcriptional modification sites from RNA sequences. *Brief Bioinform*, 2020. 21(5): p. 1676-1696.
2. Luo, X., et al., RMVar: an updated database of functional variants involved in RNA modifications. *Nucleic Acids Res*, 2021. 49(D1): p. D1405-D1412.
3. Chen, K., et al., RMDisease: a database of genetic variants that affect RNA modifications, with implications for epitranscriptome pathogenesis. *Nucleic Acids Res*, 2021. 49(D1): p. D1396-D1404.

Reviewer #3 (Remarks to the Author):

In this manuscript, Luo and Ke develop a robust deep learning method to identify sites of m6A methylation. Using this method, the authors model cis-regulatory sequences that may influence m6A deposition. They find that such cis-regulatory elements are mostly present in the 50 nucleotides downstream of the m6A site. Further, the authors find that synonymous mutations can affect m6A modification and the TGA stop codon favors m6A deposition. These models are supported by genetic variation and evolutionary conservation.

Overall, this is a VERY strong manuscript that begins to tackle key questions in the m6A field. In my opinion, one of the biggest unknowns in this field is why some m6A motifs get modified and others don't. The data presented here offer intriguing insights into how this happens. However, more analyses are required to strengthen the manuscript and corroborate key points made.

Major points:

1. The finding that the main enhancer sequences downstream of m6A sites are other m6A motifs is very intriguing and conceptually makes sense! However:

a. How often are miCLIP/m6A-CLIP sites found within 50 nt of each other? This should be discussed further. Is there a correlation between peak height in m6A-seq/MeRIP-seq and number of RRACH motifs?

b. Given that m6A enhancers mostly include parts of the m6A motif itself, it stands to reason that these motifs may also be modified. Are these adjacent enhancer motifs also called as high confidence m6A sites by the iM6A? The authors should describe this as a separate figure panel.

c. How does the last m6A motif in such a cluster of m6A motifs compare to the first in terms of its modification probability as called by the model?

d. An alternative/complementary hypothesis is that these m6A motifs serve to recruit and retain the MTC within the relevant region for modification. Can data from existing MTC PAR-CLIP/iCLIP/CLIP-seq datasets be used to infer whether such regions with higher concentrations of m6A motifs interact more with the MTC?

2. The use of existing datasets which couple m6A-profiled primary cells from 60 donors, as well as using existing genetic variation to test the veracity of iM6A-derived enhancer and silencer motifs in the regions adjacent to m6A sites is innovative and convincing. However, these data can be bolstered by molecular biology data that proves the point further. The authors should generate reporters with different cis-elements downstream of m6A motifs and test the methylation of the target sites by a method such as SCARLET or MazF-qPCR?

3. A major open question in the m6A field is whether and if so, how, m6A modification is altered by stimulation. Would the cis-elements identified to affect m6A deposition change under the context of cellular stress or perturbation. The authors should perform similar analyses as presented in the initial figures using existing miCLIP/m6A-CLIP datasets that also include a perturbation condition. At least one such dataset is that generated by Meyer et al (PMID: 26593424). Alternative analyses that explore the concept of stimulation changing (or not) m6A cis-regulatory elements will also be fine!

4. The analysis that swaps human and mouse models to demonstrate a shared m6A code through evolution is fantastic. But how far back does this go? Several miCLIP datasets now exist for *Drosophila*

(PMID: 28675155, 33674589). Perhaps the authors can train iM6A for *Drosophila* and test whether the shared m6A code exists in invertebrates as well.

Minor points:

1. The presented example of rs760539449 A->G in the *SOX10* gene also disrupts an m6A motif directly. Was this m6A motif called as being methylated by iM6A?
2. The Zenodo link cited in the end where the data is deposited is not yet easily available and requires extra permissions than merely signing up.

Reviewer #1 (Remarks to the Author):

This paper by Luo and Ke presents a new approach to sequence-based m⁶A site prediction and analysis with helpful comparisons between species and to sites identified using various sequencing approaches. However, as noted in the introduction, other m⁶A site predictors have been developed. At this point, the locations of m⁶A have been well-characterized (or at least often characterized) in both human and mouse transcriptomes. Of note this year, Körtel et al. (2021) developed an updated protocol for miCLIP and used this data in combination with a machine learning approach (also with an AUC of 0.99) to identify m⁶A sites in four human and mouse cell lines. It is unclear how iM6A compares, and biological insights from sequence modifications seem to primarily capture the known importance of RRACH motifs.

We appreciate Reviewer #1's valuable comments for improving our manuscript. We have carefully revised the manuscript accordingly. Below is our point-to-point response. The reviewer's comments are in blue, our responses are in black.

Körtel et al. (2021) developed a machine learning approach (m⁶Aboost) to identify m⁶A sites. It's noteworthy that this is a tool to identify m⁶A CLIP sites from m⁶A peaks (i.e. identifying the m⁶A-CLIP sites from the antibody non-specific binding sites), not a predictor to predict m⁶A sites in whole transcriptome using primary sequence alone.

Although several machine-learning or deep-learning methods have been developed to predict m⁶A sites, how m⁶A deposition achieves site-specificity is unknown. Our iM6A work is the first paper to reveal that this site-specificity is determined by primary nucleotide sequence, and the *cis*-elements within 50nt downstream region regulate m⁶A deposition.

1. Do negative training sites center on adenosine residues and RAC or RRACH motifs?

Thanks for the question of training dataset, the negative training sites were not centered on A residues of RAC or RRACH motif. As we described in the methods part of model training and testing, we used the pre-mRNA sequence as input. For the nucleotides in a transcript, the m⁶A sites were served as positive sites, while all other nucleotides (non-m⁶A A, G, T and C) were used as negative sites, and the probability score for each nucleotide being a m⁶A site was calculated.

2. If using 5000 nucleotides on either side of pre-mRNA as training input, how are sites encoded that don't have 5000 bases on both sides? Is this information used by the model and could it bias the models towards the prediction of m⁶A sites in the last exon?

Thanks for the question. As we described in the Methods ("*The training input of iM6A for each gene is the full length of the pre-mRNA sequence with 5000 nucleotides on each side*"), it means that the input sequence covers the region from 5kb upstream of TSS (transcription start site) to 5kb downstream of TES (transcription end site) for each gene. This strategy guaranteed that each nucleotide in pre-mRNA has at least 5000 nucleotides on both sides. Our model did not show any bias towards the prediction of m⁶A sites in last exon. The m⁶A sites in any position of pre-mRNA can be predicted. As we described in the manuscript, the same *cis*-element rule governs m⁶A deposition in both last exon and long internal exon (**Fig. 2g**).

3. Körtel et al. claim that their method of filtering training data for positive and negative sites offers improvements over standard m⁶A-CLIP/miCLIP analysis methods. Does using their sites to train iM6A change predictions (i.e. how sensitive is the method to differences in training data)?

Thanks for the suggestion. As Reviewer#1 mentioned, the m⁶Aboost in Körtel et al (2021) is a method to identify m⁶A CLIP sites from m⁶A peaks and it's not a predictor to predict m⁶A sites based on primary nucleotide sequence. We also tested the sites generated by miCLIP2, and iM6A could precisely predict the identified experimental sites. Thanks to Reviewer #1's suggestion, we now include this experimental data as an additional experimental validation to our iM6A modeling. (**Extended Data Fig. 1h, i, k**)

4. Based on their miCLIP2 data, Körtel et al. predict many non-DRACH motif sites as methylated. What percentage of these are captured by the iM6A model? Does iM6A predict any non-RAC motifs as methylated?

Thanks for the suggestion. It's known that the m⁶A site consensus is RRACH or RAC in the mRNAs of mouse and human (PMID: 26404942). In the data of miCLIP2, only 217 non-RAC motif sites were identified as very rare events (< 2% of total CLIP sites). As we described in

the manuscript, we trained iM6A with m⁶A sites in RAC motifs as positive sites. Non-RAC motifs could not be predicted as being methylated because of their extremely low probability.

5. Fig 1d: in comparison to MAZTER-seq and other methods, rather than considering the rank of sites, what percent of sites experimentally detected as methylated were predicted to be methylated by iM6A? Can you show a scatter plot of rank vs. probability for these sites (and a distribution of probabilities for tested sites overall)? How do probabilities from iM6A compare to methylation rates per site quantified using MAZTER-seq?

Thanks for the suggestion. We have showed the modeled probability by iM6A agreed with the m⁶A methylation level quantified by either m⁶A-seq or MAZTER-seq (**Extended Data Fig1m, n and o**). Even though the modeled probability has a strong correlation with the methylation level, iM6A is a method to predict the probability of a site being a m⁶A site, not to predict the methylation level of a site. To address Reviewer#1's question, the scatter plot of modeled probability versus cleavage efficiency changes by MAZTER-seq was showed and a strong correlation could be observed between the two values (**Response Figure 1A**).

Response Figure 1. (A) Scatter plot of modeled probability versus cleavage efficiency changes by MAZTER-seq for m⁶A sites. The R value was calculated by Pearson Correlation Coefficient.

6. Fig 2: Figure 2h suggests that the top enhancer motifs frequently encompass the methylated residue. As RRACH motifs are already widely recognized to be important for methylation, how does excluding sites directly surrounding the methylated adenosine

change the “enhancer” and “silencer” sets? If RRACH motifs are still among the top enhancers, would this be because they are themselves methylated? Or could this be an artifact of multiple m⁶A sites in close proximity increasing peak enrichment in miCLIP data without necessarily affecting one another’s methylation rates?

First, the motif analysis in our manuscript did exclude sites directly surrounding the methylated adenosine, we have described the details in the Methods (“For each position in the downstream region of a m⁶A site (i.e. from position 3 to position 46), the nucleotide was substituted by each of three other nucleotides. The resulted probability change (Δ Probability) of this m⁶A site is calculated by iM6A.”). Second, whether enhancer motifs being methylated is a very interesting question, and Reviewer #3 asked a related question. We plotted the distribution of RAC sites flanking the m⁶A sites. The RAC sites adjacent to m⁶A sites have a higher frequency to be m⁶A sites (**Response Figure 3C**), indicating it’s more likely to be methylated. The RAC sites adjacent to non-m⁶A sites have lower frequency to be m⁶A sites (**Response Figure 3D**), indicating it’s unlikely to be methylated. Moreover, both methylated and non-methylated RAC sites are enriched in the downstream 50 nt region of m⁶A site (**Response Figure 3C**), indicating both could enhance m⁶A deposition. Thanks to Reviewer#1 and #3’s suggestion, we now include these two panels as **Extended Data Fig. 2m,n** in manuscript, and the text was modified accordingly.

7. (Lines 345-346) Do the SNVs discussed as putative factors in disease through changes in m⁶A lead to synonymous mutations? Can effects from changes at the epitranscriptomic level be separated from sequence changes at the protein level? In this section of the text in general, do any of the SNVs detected lead to synonymous mutations?

Thanks for the suggestion. To focus on the effect of SNVs on RNA not protein level, we identified out the SNVs that only cause synonymous mutations. Many of these SNVs could affect m⁶A deposition, either enhancing or dampening (**Response Figure 1B**), and the events that could change the m⁶A probability ($|\Delta$ Probability| \geq 0.1) were also highly enriched in the region 50 nt downstream of the m⁶A sites (**Response Figure 1C**), agreeing with the finding in **Fig. 2 and Fig. 4b,c**. Altogether, iM6A could annotate the synonymous SNVs that can affect m⁶A deposition. We have added this figure as **Extended Data Fig. 4** in the manuscript.

Response Figure 1. (B) Scatter plot of predicted probability for m⁶A sites with major allele (ProbREF) or minor allele (ProbALT), all SNVs are synonymous mutations. Red color dots were mutational events that increased m⁶A probability ($\Delta\text{Probability} \geq 0.1$); Green color dots are mutational events that decreased m⁶A probability ($\Delta\text{Probability} \leq -0.1$). **(C)** Positional plot of $\Delta\text{Probability}$ (cutoff = 0.1) for m⁶A sites with major allele or minor allele, all SNVs are synonymous mutations. Red color dots were mutational events that increased m⁶A probability; Green color dots were mutational events that decreased m⁶A probability.

8. In Fig 3f/h & 4e/g, examples focus on direct changes to the RRACH motif encompassing methylated sites. It is not surprising that these would affect methylation rates. The same is true in Figure 6, focusing on stop codons. Again, stop codons seem to be important mainly as part of RRACH motifs or their direct extensions, with differences falling off within 3 nucleotides of methylated sites (Figure 6a). Do conclusions regarding stop codon associations with m⁶A represent anything beyond the likelihood of various stop codons to be included in or to disrupt favorable RRACH motifs? (Lines 427 & 430): it seems particularly obvious that a change in the stop codon at position 2 that changes A to G would abrogate m⁶A methylation at that A (there is no longer an adenosine to be methylated), while a change from G to A would have the opposite effect.

Thanks for the advice. While Fig 3f/h & 4e/g were positive controls for iM⁶A prediction accuracy, Figure 3g/i & 4f/h showed how the downstream SNVs affect m⁶A deposition. For the m⁶A site adjacent to stop codon, the m⁶A motif of sites at position -2 (i.e. NRACTRR. NRACN is the motif of m⁶A and TRR is the stop codon) was not affected by stop codon swap (**Fig 6a, 6b**). The TGA stop codon is still in favor of m⁶A deposition.

9. Regarding Figure 7, could errors when applying cross-species model reveal anything about species-specific m⁶A deposition preferences?

Thanks for the suggestion. According to the data of this manuscript, the human and mouse have the similar rules governing m⁶A deposition. Same as the question 4 of Review #3, we are also interested in how m⁶A deposition is determined in different species across vertebrates and invertebrates. However, the training of iM6A needs a large number of high-quality experimentally determined m⁶A sites as positive sites. The currently published sites of different species could not satisfy the requirement, and we plan to investigate this question in the future when these data become available.

10. If it is true that sequence changes downstream of m⁶A sites have a greater effect on m⁶A probability than sequence changes upstream, any predictions as to mechanism?

This is a very interesting question, and Reviewer #3 asked a related question. We analyzed the ChIP-seq data of METTL3 (PMID: 28581511, PMID: 32778823), and found METTL3 is enriched in the downstream region of m⁶A sites (**Response Figure 3F, 3G**). We hypothesized that the methyltransferase complex (MTC) might play a role in this mechanism. The precise mechanism is still yet to be established and needs a lot of focused research investigations as one of the future directions.

11. Given that many previously published m⁶A site predictors also claim high accuracy, could the same biological insights have been generated based on these tools rather than iM6A? What is the overlap in sites predicted using these different methods?

As we showed in **Fig 1b**, the performance of iM6A is much better than any of the predictors published previously. High accuracy is the foundation of biological discovery. Based on iM6A modeling, we could understand the cis-element mechanism for m⁶A site-specificity.

12. m⁶A methylation rates are reported to differ across cell lines (e.g. Liu et al., 2013 SCARLET paper). Does accuracy decrease if iM6A is trained on data from one cell line and tested on data from another?

This is a very interesting point. While the methylation rates of a m⁶A site may be different across cell lines, the iM6A is a deep learning model to predict the probability of a site being a m⁶A site but not a tool to predict the methylation level, though the probability modeled by iM6A has a strong correlation with the methylation level. At the same time, a high-throughput method for quantifying genome-wide m⁶A site methylation rate is yet to be developed for different cell lines. We hope to further develop deep learning models for this methylation rate modeling once the related technology becomes available in the future.

Minor comments:

- Table S2: Is the p-value really exactly 0 for all pentamers?

The p-value is less than 2.2250738585072014E-308, and python cannot show the value less than 2.2250738585072014E-308. Moreover, the minimum value of Excel is 2.00E-308.

Accordingly, we modified the p-value as less than 5.00E-308 in Table S2.

- Ext Fig 3: a lot of text in the legend is repeated for each subfigure and could be condensed.

Thanks for the advice, we have modified the figure legends in the manuscript.

Reviewer #2

The manuscript presented a deep learning model for m⁶A site prediction and interpretation. In general, the manuscript is well written, the experiments were well designed with multiple different technical considerations.

The modeling of m⁶A sites using whole Pre-mRNA is indeed novel and can potentially bring major improvement in performance. Many experiments were designed to show the performance of the predictor and great efforts were made to interpret the model and its biological meaning. I did see great efforts of the authors and truly appreciate that.

We appreciate Reviewer#2's high recognition of our work and valuable comments. We have carefully revised the manuscript according to the comments which improved our manuscript. Below is our point-to-point response. The reviewer's comments are in blue, our responses are in black.

1. Truly sorry for being skeptical. The reported performance (AUROC=0.99) seems too good to be true based on existing literature [1]. This is due to the highly imbalanced dataset, with a lot more negative sites than positive sites. AUROC is known to be strongly affected by data imbalance. It is better to calculate the AUPRC, which is not effected by sample imbalance, and also report the AUROC on balanced data (with the same number positive and negative sites or samples). Ideally, the same number of negative and positive sites should be extracted from the same transcript as in most existing works for RNA modification site prediction.

As Review #2 described, many existing publications (e.g. PMID: 31714956) for m⁶A sites prediction are based on the sequence flanking the target sites (m⁶A or non-m⁶A sites), and a 1:1 ratio of positive-to-negative sites were extracted from the same transcript.

Since m⁶A is deposited to nascent pre-mRNA (PMID: 28637692), we modeled how m⁶A deposition is determined by pre-mRNA primary sequence through the iM6A deep learning modeling. As we described in the methods part of the model training and testing, we used the pre-mRNA sequence as input. For the nucleotides in a transcript, the m⁶A sites were served as positive sites, while all other nucleotides (non-m⁶A A, G, T and C) were used as negative sites, and the probability score for each nucleotide being a m⁶A site was calculated. It's a remarkable fact that the negative sites contribute useful primary sequence information same as the positive sites, and both positive and negative sites contribute to the modeling performance. It is an important advantage of the iM6A deep learning modeling that it can intake huge amount of information including both positive and negative sites.

To address reviewer's question, we calculated the AUPRC for the independent testing dataset, which was showed in **Supplementary Table1** and AUPRC=0.45. To assess the performance of iM6A's prediction for the RRACH sites on chromosome 9 (the independent testing dataset), we calculated the AUPRC value which showed iM6A had much better performance than SVM or CNN-RNN model (**Extended Data Fig1b, 1c**). Thus, in summary, both AUROC and AUPRC values showed that iM6A has much better m⁶A deposition modeling than the existing methods. Thanks to Reviewer #2's comments, we now include all these new figures into our manuscript.

2. “pre-mRNA” has been used to describe the input sequence. Do you also consider lncRNA? Can the negative sites be from intronic regions, which are often easier to recognize and leads to higher reported performance?

We only used the protein-coding genes in this study, while the long non-coding RNAs (lncRNA) are not included. Although some lncRNAs such as *Malat1* have m⁶A sites in their transcripts, most of m⁶A sites are enriched in mRNAs (PMID: 26404942, PMID: 28637692).

The input sequence covers the transcripts from 5kb upstream of TSS (transcription start site) to 5kb downstream of TES (transcription end site). The intronic regions are also included in this study, which are part of the transcript structure and essential for modeling m⁶A deposition on pre-mRNA.

For the performance of m⁶A-centered models, the full transcript model (i.e. with intron) could achieve better performance than mature mRNA model (PMID: 30993345). By excluding the RRACH sites in intronic region, we calculated the performance of iM6A’s prediction for the RRACH sites only in the exonic regions, and the AUROC is 0.921 (**Response Figure 2A**), which also achieved state-of-the-art performance in comparison to the existing methods.

Response Figure 2A. Receiver operator curves (ROCs) and the corresponding area under receiver operator curves (AUROC) scores for iM6A, CNN-RNN, and SVM. Here the RRACH sites in the exonic region of mouse chromosome 9 were used to test the models.

3. Figure 1b is not accurate. WHISTLE achieved good performance because it also takes advantages of genomic features. If you use only SVM and sequence information, then it is just MethyRNA method.

Thanks for pointing out this mistake in the manuscript, we only used the sequence information in the SVM model. We've modified the text accordingly. "Receiver operator curves (ROCs) and corresponding area under receiver operator curves (AUROC) scores of iM6A, CNN-RNN (implemented in TDm6A), and SVM (implemented in MethyRNA)."

4. Authors showed a potential function of iM6A to identify the m6A associated SNVs (Figure 4). There have already been quite a few works in this field, which should probably be mentioned, including at least [2, 3].

Thanks for the suggestion. In our manuscript, we assessed how the SNVs could affect m⁶A deposition by iM6A. The existing publications include RMvar and RMDisease that collected the SNVs being potentially involved in m⁶A modification. iM6A can systematically predict how m⁶A deposition could be influenced by the SNVs and could provide synergistic contribution to the scientific community in addition to RMvar and RMDisease. We have discussed and cited these publications in our manuscript, and the text was also modified accordingly. "Defining the disease-associated mutations among millions of SNVs is a grand challenge. The databases like RMvar, RMDisease collected the genetic variants which might be associated with m⁶A modification, while iM6A could provide synergistic contribution to decipher the cis-element mechanisms and could provide a new perspective in understanding the diseases caused by RNA modifications."

5. Please provide more details related to the encoding of m6A sites. In the METHOD section. Authors described that 'The training input of iM6A for each gene is the full length of the pre-mRNA sequence with 5000 nucleotides on each side. This means the input are not of the same dimension for different genes. How do you deal with super long transcripts (100 thousands nt)? Do you consider pre-mRNA only, or you also considered pre-lncRNA. How do you label the m6A sites in the encoded information (it should be part of the input, and not clear to me based on Figure 1a or Figure 6b)?

As we described in the METHOD ("The training input of iM6A for each gene is the full length of the pre-mRNA sequence with 5000 nucleotides on each side"), it means the input sequence covers the region from 5kb upstream of TSS to 5kb downstream of TES for each

gene. The sequence was transformed by One-hot-Encoding. Then, the one-hot encoded nucleotide sequence was split into blocks of length 5000+5000+5000 in such a way that the i^{th} block consisted of the nucleotide sequence position from $5000(i-1)-5000+1$ to $5000i+5000$. Similarly, the output label was also split into blocks of length 5000 in such a way that i^{th} block consists of the positions from $5000(i-1) + 1$ to $5000i$. This strategy was also adopted by SpliceAI (PMID: 30661751) to model pre-RNA splicing. We have modified the text of model architecture in Methods accordingly.

In this study, we only focused on pre-mRNA, as we answered in the response to Question 2.

6. Please provide your training and testing data as well, so that people can fully replicate the reported performance. (Apologize for being skeptical. But I am sure that you don't want to find out later that the high performance is due to unfair setting or a mistake.)

Thanks for the suggestion, we've uploaded the datasets to Github (<https://github.com/ke-laboratory/iM6A>). The whole dataset was divided into training and testing datasets. The training dataset contained all the transcripts on most chromosomes except chromosome 9 (chr9), the transcripts of which were held out and reserved for the testing. We documented this detail in the Methods section of model training and testing.

7. If the *cis*-elements regulating the m⁶A deposition preferentially reside within the 50 nt downstream of the m⁶A sites, why the input sequence of your model is so long (at least 10kb)? Most existing method uses only 1kb sequences. Could you test the performance of your method with shorter sequences on both sides?

Even though this study showed the *cis*-elements regulating m⁶A deposition largely reside within the 50nt downstream of the m⁶A sites, other sequence features could also affect m⁶A deposition as long-range regulations, which are interesting future directions that we are working on.

We were also interested in the performance of iM6A with shorter sequences on both sides. We trained the iM6A model with 80, 400, 2K, and 10K sequence on both sides, and the performance increased along with sequence length. (**Response Figure 2B**).

Response Figure 2B. Receiver operator curves (ROCs) and corresponding area under receiver operator curves (AUROC) scores of iM6A with different sequence length. Here mouse chromosome 9 data was used to test the iM6A, which were trained independently on data of other mouse chromosomes except chromosome 9.

8. Sorry for my ignorance. How do you define “m6A enhancers” and “m6A silencers”? This is the first time I saw these two terms.

The pentamers that could enhance m⁶A deposition are designated as m⁶A enhancers, while the pentamers that could dampen m⁶A deposition are designated as m⁶A silencers. The same terms have been used to study regulatory *cis*-elements of pre-mRNA splicing (e.g. PMID: 21659425).

Reviewer #3:

In this manuscript, Luo and Ke develop a robust deep learning method to identify sites of m6A methylation. Using this method, the authors model *cis*-regulatory sequences that may influence m6A deposition. They find that such *cis*-regulatory elements are mostly present in the 50 nucleotides downstream of the m6A site. Further, the authors find that synonymous mutations can affect m6A modification and the TGA stop codon favors m6A deposition. These models are supported by genetic variation and evolutionary conservation.

Overall, this is a VERY strong manuscript that begins to tackle key questions in the m⁶A field. In my opinion, one of the biggest unknowns in this field is why some m⁶A motifs get modified and others don't. The data presented here offer intriguing insights into how this happens. However, more analyses are required to strengthen the manuscript and corroborate key points made.

We highly appreciated Reviewer #3's recognition of our work. We modeled m⁶A deposition by deep learning and found the *cis*-elements determine its site-specificity. Moreover, Reviewer #3's valuable comments improved our manuscript. We have carefully revised the manuscript according to the comments. Below is our point-to-point response. The reviewer's comments are in blue, our responses are in black.

Major points:

1. The finding that the main enhancer sequences downstream of m⁶A sites are other m⁶A motifs is very intriguing and conceptually makes sense! However:

a. How often are miCLIP/m⁶A-CLIP sites found within 50 nt of each other? This should be discussed further. Is there a correlation between peak height in m⁶A-seq/MeRIP-seq and number of RRACH motifs?

This is a very interesting point. We clustered the miCLIP/m⁶A-CLIP sites within 50 nt of each other. ~17000 sites are single individual sites, while over 34000 sites could be grouped as clusters (**Response Figure 3A**). We also measured the correlation between the peak enrichment score and its number of RAC motifs of each peak in m⁶A-seq/MeRIP-seq, and we did see some correlation ($R=0.18$, $P < 2.2E-16$) (**Response Figure 3B**).

Response Figure 3. (A) Count of miCLIP/m⁶A-CLIP sites (in RAC motifs) in clusters. The miCLIP/m⁶A-CLIP sites were clustered into clusters (within 50 nt). Single means the cluster has only one site, while multiple means the cluster has at least two sites. **(B)** Scatter plot was between the peak enrichment score and the number of RAC sites of each peak, and each dot was a m⁶A peak identified by m⁶A-CLIP (PMID: 26404942). The peak enrichment score was calculated as the average of the peak enrichment scores (10nt interval) in a m⁶A peak. The R value was calculated by Pearson Correlation Coefficient.

b. Given that m⁶A enhancers mostly include parts of the m⁶A motif itself, it stands to reason that these motifs may also be modified. Are these adjacent enhancer motifs also called as high confidence m⁶A sites by the iM6A? The authors should describe this as a separate figure panel.

The crosstalk of m⁶A sites in cluster is a very interesting point. We plotted the distribution of RAC sites flanking the m⁶A sites. The RAC sites adjacent to m⁶A sites have a higher frequency to be m⁶A sites (**Response Figure 3C**), indicating it's more likely to be methylated. The RAC sites adjacent to non-m⁶A sites have lower frequency to be m⁶A sites (**Response Figure 3D**), indicating it's unlikely to be methylated. Moreover, both methylated and non-methylated RAC sites are enriched in the downstream 50 nt region of m⁶A sites (**Response Figure 3C**), indicating both could enhance m⁶A deposition. Thanks to Reviewer #3's suggestion, we now include these two panels as **Extended Data Fig. 2m,n** in manuscript, and the text was modified accordingly.

Response Figure 3. (C) Positional plot for the frequency of RAC sites in the sequences around the m⁶A sites. The plots were compared between the m⁶A RAC sites (red color, higher

m⁶A probability, probability ≥ 0.05) and the non-m⁶A RAC sites (control, green color, lower m⁶A probability, probability < 0.05). Data were presented as mean \pm S.E.M. (standard error of the mean). (D) Positional plot for the frequency of RAC sites in the sequences around the non-m⁶A sites. The plots were compared between the m⁶A RAC sites (red color, higher m⁶A probability, probability ≥ 0.05) and the non-m⁶A RAC sites (control, green color, lower m⁶A probability, probability < 0.05) of RAC sites. Data were presented as mean \pm S.E.M. (standard error of the mean)

c. How does the last m⁶A motif in such a cluster of m⁶A motifs compare to the first in terms of its modification probability as called by the model?

This is a very interesting question. We identified the m⁶A sites as RAC sites with high probability value (Probability ≥ 0.05), then clustered these sites within 50nt of each other.

Scatter plot showed little bias of its modification probability between the first m⁶A site and the last m⁶A site in a cluster (**Response Figure 3E**).

Response Figure 3. (E) Scatter plot of probability value between the first and the last m⁶A sites in a cluster, each dot was a m⁶A site cluster.

d. An alternative/complementary hypothesis is that these m⁶A motifs serve to recruit and retain the MTC within the relevant region for modification. Can data from existing MTC PAR-CLIP/iCLIP/CLIP-seq datasets be used to infer whether such regions with higher concentrations of m⁶A motifs interact more with the MTC?

This is a very interesting question, and Reviewer #1 asked a related question. Though there is no good quality METTL3-CLIP data publicly available, we analyzed the ChIP-seq data of METTL3 (PMID: 28581511, PMID: 32778823), and found METTL3 is likely to be enriched in the downstream region of m⁶A sites (**Response Figure 3F, 3G**). We hypothesized that the methyltransferase complex (MTC) may play the role in this mechanism. The precise mechanism is yet to be established, and we are working on this question as one of the future directions.

Response Figure 3. (F-G) The mouse (F) or human (G) METTL3 ChIP-seq peak density in last exons were compared between m⁶A sites and non-m⁶A sites. The peak density was calculated as the number of METTL3 peak regions per 10-nt interval, and the raw ChIP-seq data is from PMID: 28581511 & PMID: 32778823.

2. The use of existing datasets which couple m⁶A-profiled primary cells from 60 donors, as well as using existing genetic variation to test the veracity of iM⁶A-derived enhancer and silencer motifs in the regions adjacent to m⁶A sites is innovative and convincing. However, these data can be bolstered by molecular biology data that proves the point further. The authors should generate reporters with different cis-elements downstream of m⁶A motifs and test the methylation of the target sites by a method such as SCARLET or MazF-qPCR?

We appreciate Reviewer#3's comments that our work using existing genetic variation to test iM⁶A-derived enhancer and silencer motifs as being innovative and convincing. We also agree with Reviewer#3's suggestion that molecular biology data could further prove the point. We do have plans in this direction combining individual reporters with high-throughput ones to systematically test the cis-element rules of m⁶A deposition, however, this project would constitute more appropriately as a separate paper for its scope and the time/efforts to be

devoted, particularly considering that our current iM6A work is already a full-size article of seven main figures (each with many panels) plus many supplemental figure panels. Here we really hope to have Reviewer#3's support to let us publish the iM6A work as it is right now with no more delay. The sooner we publish this iM6A work, the sooner we could share our important finding with the scientific community. We appreciate a lot this support, and thank you.

3. A major open question in the m6A field is whether and if so, how, m6A modification is altered by stimulation. Would the cis-elements identified to affect m6A deposition change under the context of cellular stress or perturbation. The authors should perform similar analyses as presented in the initial figures using existing miCLIP/m6A-CLIP datasets that also include a perturbation condition. At least one such dataset is that generated by Meyer et al (PMID: 26593424). Alternative analyses that explore the concept of stimulation changing (or not) m6A cis-regulatory elements will also be fine!

Thanks for the suggestion. It's a very interesting topic how m⁶A deposition is altered by stimulation. The dataset generated by Meyer et al (PMID: 26593424) did not have the m⁶A sites with single nucleotide resolution under heat shock, thus we could not train deep learning model. We are definitely very interested in pursuing this direction as future projects once the related data set becomes available.

4. The analysis that swaps human and mouse models to demonstrate a shared m6A code through evolution is fantastic. But how far back does this go? Several miCLIP datasets now exist for *Drosophila* (PMID: 28675155, 33674589). Perhaps the authors can train iM6A for *Drosophila* and test whether the shared m6A code exists in invertebrates as well.

Thanks for the suggestion. Same as the question 9 of Review #1, we are also interested in how m⁶A deposition is determined in different species beyond human and mouse. However, the training of iM6A needs a large number of high-quality experimentally determined m⁶A sites as positive sites. The two published m⁶A miCLIP sites in *Drosophila* (PMID: 28675155, 33674589) could not meet this requirement (personal communication with Dr. Eric C. Lai and

his team). We would need to work on this direction as future projects through a potential collaboration with Dr. Eric C. Lai Lab.

Minor points:

1. The presented example of rs760539449 A->G in the SOX10 gene also disrupts an m6A motif directly. Was this m6A motif called as being methylated by iM6A?

Yes, the probability value of this site is 0.56. It's a m⁶A site called by iM6A.

2. The Zenodo link cited in the end where the data is deposited is not yet easily available and requires extra permissions than merely signing up.

We are truly sorry for the inconvenience of accessing the data. We have updated the Zenodo link (<https://zenodo.org/record/4734266>), and now it is fully accessible.

REVIEWER COMMENTS

Reviewer #1 (Remarks to the Author):

Thank you to the authors for their responses and additional analyses. I have a few questions about the model and conclusions remaining.

1. It seems like both Reviewer 2 (Question 5) and I had the similar questions regarding the upstream and downstream +/- 5000 bases included in the model. The confusion seems to stem from the authors' phrasing regarding the use of pre-mRNA sequences. If I understand correctly now, the authors in fact also consider the genomic sequence that surrounds the pre-mRNA +/- 5000 bases. This should be clarified and specifically noted in the methods. But this clarification also begs the follow-up questions: what biological contribution, if any, could the intergenic sequence surrounding a transcript have on m6A deposition? If none, what is that data contributing to the model?

Can the authors also please elaborate on the statement in their response that "Our model did not show any bias towards the prediction of m6A sites in last exon"? As they note in the text (e.g. lines 59-61), it would be expected based on previous literature that m6A would be enriched in the 3' UTR and last exon. A metagene plot showing enrichment of predicted methylation over normalized lengths of 5' UTR, CDS, 3' UTR would clarify whether the distribution predicted by the model is as expected. In theory, though, if this enrichment does exist in the last exon, differences in sequence composition between pre-mRNA and the following intergenic region could contribute to predictions. That would bias the model towards prediction of m6A in the last exon using information beyond that contained in the pre-mRNA sequence theoretically available to a methyltransferase complex (this comes back to the question above regarding the biological justification for inclusion of intergenic sequences in the model). One way of checking for this bias would be to compare accuracy rates for internal sites, where no intergenic sequence was included in the model, and distal sites within 5000 bp of the TES, where the intergenic sequence could contribute to predictions.

The claims of the authors focus on regulatory elements within 50 bp of methylation sites, and later analyses specifically select for "m6A sites which located at least 50 nt away from both last exon start and last exon end" (lines 846-847). The comparison between sites in the last exon and in "long internal exons" (although how these long exons are defined needs to be described in the methods section) in Figure 2 also supports a role for regulatory pentamers independent of location in a gene. I therefore wouldn't expect contributions from intergenic sequences to invalidate the main conclusions of the paper, but they could help explain why training with extensive sequences substantially increases the accuracy of this model compared to 80 nt windows, as shown in Response Figure 2B (an important figure that should be added to the supplementary information).

2. To rephrase part my question regarding the stop codon analysis, are the motif preferences demonstrated a property of stop codons, or simply of their motifs (is TGA generally associated with m6A in surrounding RRACH motifs, or only TGA that codes for stop codons)? If the authors check the same

three trimers in locations that don't surround the stop codon for association with m6A, do the same trends hold? This wouldn't change that a particular stop codon may be more associated with m6A than another, however, a general motif preference could help explain why.

3. Regarding SNVs: "We further categorized the SNVs" (line 356): do "the SNVs" refer to SNVs that cause synonymous mutations, described in the previous sentence, or to a larger group of SNVs again?

Could the authors please clarify whether the SNVs they focus on in Figure 4e-h are synonymous or not? The claim that, for instance, "The two SNVs above could affect the m6A modification in the DARS2 transcript as a novel disease cause" (lines 370-371) may not be the most likely hypothesis if those SNVs also alter protein sequence.

The association between changes in m6A deposition and increased pathogenicity in general seems questionable. To better show the association, rather than showing a jitter plot in Figure 4a (where the distributions are difficult to see among overlapping points) and the barplot in Figure 4d (that binarizes sites into m6A changed or not changed based on an arbitrary threshold), could the authors should a histogram or density plot to show the distribution of m6A probability changes for the three categories of SNV? These distributions do not appear different in Figure 4a, but the claims surrounding Figure 4d would suggest that they are.

Reviewer #2 (Remarks to the Author):

Glad to confirm that all my comments have been properly addressed. I am happy to recommend the acceptance of this manuscript at NC.

Reviewer #3 (Remarks to the Author):

This reviewer's concerns have been satisfied. I agree that several points raised by reviewers would benefit from detailed analyses in future publications. As such, the manuscript in its current form is a strong and important piece of work and will greatly contribute to our understanding of the details underlying m6A modification.

Reviewer #1 (Remarks to the Author):

Thank you to the authors for their responses and additional analyses. I have a few questions about the model and conclusions remaining.

We appreciate the valuable comments from the reviewer to improve the manuscript. Below is our point-to-point response. The reviewer's comments are in blue, our responses are in black.

1. It seems like both Reviewer 2 (Question 5) and I had the similar questions regarding the upstream and downstream +/- 5000 bases included in the model. The confusion seems to stem from the authors' phrasing regarding the use of pre-mRNA sequences. If I understand correctly now, the authors in fact also consider the genomic sequence that surrounds the pre-mRNA +/- 5000 bases. This should be clarified and specifically noted in the methods. But this clarification also begs the follow-up questions: what biological contribution, if any, could the intergenic sequence surrounding a transcript have on m⁶A deposition? If none, what is that data contributing to the model?

We apologize for the misunderstanding of the input sequence. We have modified it in the methods. (*The training input of iM6A for each gene is the full length of the pre-mRNA sequence with 5000 nucleotides on each side, covering the transcript from 5kb upstream of TSS (transcription start site) to 5kb downstream of TES (transcription end site).*) It is known that transcriptional initiation and termination could regulate pre-mRNA processing event such as pre-mRNA splicing and poly-adenylation (PMID: 24514444 and PMID: 27677860). As m⁶A mRNA modification is deposited to pre-mRNA during transcription (PMID: 28637692), it could be true that transcriptional initiation and termination may also regulate m⁶A deposition along with other pre-RNA processing events, thus providing the biological rational to include the 5000 bases flanking the pre-mRNA. We are interested in the biological role of regulatory regions like promoter and transcript termination in m⁶A deposition, and would like to work on it as one of the future directions.

Can the authors also please elaborate on the statement in their response that "Our model did

not show any bias towards the prediction of m⁶A sites in last exon”? As they note in the text (e.g. lines 59-61), it would be expected based on previous literature that m⁶A would be enriched in the 3' UTR and last exon. A metagene plot showing enrichment of predicted methylation over normalized lengths of 5' UTR, CDS, 3' UTR would clarify whether the distribution predicted by the model is as expected. In theory, though, if this enrichment does exist in the last exon, differences in sequence composition between pre-mRNA and the following intergenic region could contribute to predictions. That would bias the model towards prediction of m⁶A in the last exon using information beyond that contained in the pre-mRNA sequence theoretically available to a methyltransferase complex (this comes back to the question above regarding the biological justification for inclusion of intergenic sequences in the model). One way of checking for this bias would be to compare accuracy rates for internal sites, where no intergenic sequence was included in the model, and distal sites within 5000 bp of the TES, where the intergenic sequence could contribute to predictions.

Our original sentence that “our model did not show any bias towards the prediction of m⁶A sites in last exon” was intended to say that iM6A could accurately predict the m⁶A sites in both last exon and internal exons without technical bias issue. Here we prefer to use “enrichment” instead of “bias” to describe the enrichment of m⁶A sites in last exons as to avoid such a confusion. The top 100K m⁶A sites were used for the metagene plot, and indeed enriched in the 3'-UTR as expected (**Response Figure 1A**).

We also calculated the accuracies for both internal sites (distance to TSS (transcription start site) and TES (transcription end site) is over 5000 nt) and distal sites (within 5000 nt of TSS or TES), and iM6A can accurately predict both internal and distal sites (both AUROCs > 0.95 **Response Figure 1B, C**). The full details of iM6A modeling in internal and distal sites and their associated biological mechanism of transcriptional initiation and termination on regulating m⁶A deposition are interesting research directions that we hope to work on as separate projects in the future.

Response Figure 1. (A) Metagene plot of top 100K m⁶A sites (in RRACH motifs). (B-C) Receiver operator curves (ROCs) and the corresponding area under receiver operator curves (AUROC) scores for internal sites and distal sites. The sites which located over 5000 nt from both TSS and TES were defined as internal sites, while the sites within 5000 nt of TSS or TES were defined as distal sites. Here the RRACH sites in mouse chromosome 9 were used to test the models.

The claims of the authors focus on regulatory elements within 50 bp of methylation sites, and later analyses specifically select for “m⁶A sites which located at least 50 nt away from both last exon start and last exon end” (lines 846-847). The comparison between sites in the last exon and in “long internal exons” (although how these long exons are defined needs to be described in the methods section) in Figure 2 also supports a role for regulatory pentamers independent of location in a gene. I therefore wouldn't expect contributions from intergenic sequences to invalidate the main conclusions of the paper, but they could help explain why training with extensive sequences substantially increases the accuracy of this model compared to 80 nt windows, as shown in Response Figure 2B (an important figure that should be added to the supplementary information).

Thank you to the reviewer's suggestion to clarify long internal exon definition. The length of long internal exons for motif analysis were 100 nt at least. We have added the details in the methods section. Moreover, we thank the reviewer for the suggestion to include the original panel of Response Figure 2B from the 1st revision to the supplementary figures. Now we add it as **Extended Data Fig. 1q** and modify the text in the manuscript accordingly.

2. To rephrase part my question regarding the stop codon analysis, are the motif preferences demonstrated a property of stop codons, or simply of their motifs (is TGA generally associated with m6A in surrounding RRACH motifs, or only TGA that codes for stop codons)? If the authors check the same three trimers in locations that don't surround the stop codon for association with m6A, do the same trends hold? This wouldn't change that a particular stop codon may be more associated with m6A than another, however, a general motif preference could help explain why.

We appreciate the insights from the reviewer that TGA trimer may be in favor of m⁶A deposition beyond the stop codon itself. To test the hypothesis, we counted the number of TGA, TAA, and TAG trimers in m⁶A enhancer motifs. The TGA trimer is consistently enriched comparing to TAA or TAG, supporting the hypothesis that TGA trimer promotes m⁶A deposition. (**Response Table 1**, the same conclusion regardless of top50, top100, and top150 m⁶A enhancer motifs). Accordingly, we included Response Table 1 to the manuscript as Supplemental Table 4. and added one sentence at the Stop codon section of the manuscript: *Moreover, the TGA as a trimer motif may promote m⁶A deposition in comparison to TAA and TAG trimers (Supplemental Table 4).*

	Top50	Top100	Top150
TGA	5	13	21
TAA	0	3	13
TAG	0	0	0

Response Table 1. Number of motifs containing TGA, TAA, TAG in m⁶A enhancer motifs.

3. Regarding SNVs: “We further categorized the SNVs” (line 356): do “the SNVs” refer to SNVs that cause synonymous mutations, described in the previous sentence, or to a larger group of SNVs again?

Our apology for the writing ambiguity. “the SNVs” described in line 356 belongs to the larger group of SNVs, not the SNVs that cause synonymous mutation. We have modified the text accordingly.

Could the authors please clarify whether the SNVs they focus on in Figure 4e-h are

synonymous or not? The claim that, for instance, “The two SNVs above could affect the m⁶A modification in the DARS2 transcript as a novel disease cause” (lines 370-371) may not be the most likely hypothesis if those SNVs also alter protein sequence.

We apologize for this confusion. The Figure 4e-h did not focus on synonymous mutations, and all of the SNVs showed in these four panels cause missense mutation. Figure 4e-h were used as individual examples to show that iM6A could help to annotate the effect of pathogenic SNVs on m⁶A deposition beyond protein coding sequence mutations. In ClinVar database, the missense and nonsense SNVs are more likely to be annotated as pathogenic for their convenience in inferring protein functional disruption. In other words, the pathogenic SNVs that are documented currently in ClinVar primarily focus on protein sequence disruption. Our iM6A annotation provides an alternative angle to interpret these disease-causing SNVs from the m⁶A RNA modification perspective. As the m⁶A disease research grows mature in the future, the ClinVar database could include pathological SNVs that were affected by m⁶A deposition alone and our iM6A work could promote the disease research discovery in this direction. Accordingly, we have modified our manuscript text to make it clear.

The association between changes in m⁶A deposition and increased pathogenicity in general seems questionable. To better show the association, rather than showing a jitter plot in Figure 4a (where the distributions are difficult to see among overlapping points) and the barplot in Figure 4d (that binarizes sites into m⁶A changed or not changed based on an arbitrary threshold), could the authors should a histogram or density plot to show the distribution of m⁶A probability changes for the three categories of SNV? These distributions do not appear different in Figure 4a, but the claims surrounding Figure 4d would suggest that they are.

As we showed in Figure 4b, only a small population of SNVs cause the change of m⁶A deposition ($|\Delta\text{Probability}| > 0.1$). The current Figure 4a is to show that some SNVs (though only a small proportion) could cause the evident m⁶A deposition changes. We did the histogram plot to show the distribution of m⁶A probability changes for the three categories of SNVs (**Response Figure 2**), the difference among the three groups cannot be seen as the dominant proportion

of SNVs do not affect m⁶A deposition. Thus, we would appreciate that we could stay with our current Figure 4a plot. At the same time, to help reader understands the plot better, we added one sentence in the manuscript: *though a large proportion of SNVs don't affect m⁶A deposition, some do have evident effects on m⁶A deposition (Fig. 4a).*

Response Figure 2. Histogram of the distribution of m⁶A probability changes for VUS, Benign, and Patho SNVs.

Reviewer #2 (Remarks to the Author):

Glad to confirm that all my comments have been properly addressed. I am happy to recommend the acceptance of this manuscript at NC.

We are thankful to Reviewer #2 for the recommendation of this manuscript for publication.

Reviewer #3 (Remarks to the Author):

This reviewer's concerns have been satisfied. I agree that several points raised by reviewers would benefit from detailed analyses in future publications. As such, the manuscript in its current form is a strong and important piece of work and will greatly contribute to our understanding of the details underlying m⁶A modification.

We are thankful to Reviewer #3 for the encouraging statement for our manuscript acceptance.

REVIEWERS' COMMENTS

Reviewer #1 (Remarks to the Author):

The authors have sufficiently addressed my comments.